# A high-quality daily nighttime light (HDNTL) dataset for global 600+ cities (2012-2024)

Zixuan Pei[1], Xiaolin Zhu[1,2,*], Yang Hu[3], Jin Chen[4], Xiaoyue Tan[1]

[1]Department of Land Surveying and Geo-Informatics, The Hong Kong Polytechnic University, Hong Kong, China
[2]The Hong Kong Polytechnic University Shenzhen Technology and Innovation Research Institute (Futian), Shenzhen, China
[3]Institute of Industrial Science, The University of Tokyo, Tokyo, Japan
[4]State Key Laboratory of Remote Sensing Science, Faculty of Geographical Science, Beijing Normal University, Beijing, 100875, China

*Correspondence to*: Xiaolin Zhu (xiaolin.zhu@polyu.edu.hk)

**Abstract.** Nighttime light (NTL) data at daily scales presents an innovative foundation for monitoring human activities, offering vast potential across various research domains such as urban planning and management, disaster monitoring, and energy consumption. The VNP46A2 dataset, sourced from NPP/VIIRS, has been providing globally corrected daily NTL data since 2012. However, persistent challenges, such as fluctuations in daily NTL series due to spatial mismatch and angular effects, as well as missing data holes, have significantly impacted the accuracy and comprehensiveness of extracting daily NTL changes. To address these challenges, a dataset production framework focusing on error correction, interpolation, and validation was developed. This framework led to the creation of a high-quality daily NTL dataset from 2012 to 2024, named HDNTL, which specifically targets 653 cities with populations predictably exceeding one million in 2025. A comparative analysis with the VNP46A2 dataset revealed promising results in spatial mismatch correction for two sample areas—the airport and highway (angular effect can be ignored). These areas exhibited reduced fluctuations in HDNTL time series and enhanced spatial consistency among pixels with homogeneous light sources. Furthermore, the correction of angular effects across various urban building landscapes demonstrated sound improvements, mitigating angular effects in different directions and reducing periodicity from the angular impacts. The spatiotemporal interpolation of missing data holes shows high similarity with the reference data, as indicated by a Pearson correlation coefficient (r) of 0.99, and it increased the valid pixels of all cities by about 2%. The HDNTL dataset exhibited enhanced consistency with high-resolution SDGSAT-1 data regarding the NTL change rate and alignment with ground truth data of power outages, showcasing superior performance in short-event detection. Overall, the HDNTL dataset effectively mitigates instability in daily series caused by spatial mismatch and angular effects observed in VNP46A2, improving data comparability across time and space dimensions. This dataset enhances the ability of the NTL to reflect the ground events, providing a more accurate reference for daily-scale nighttime light research. Additionally, the dataset production framework facilitates easy updates from future VNP46A2 products to HDNTL. The HDNTL is openly available at https://doi.org/10.5281/zenodo.17079409 (Pei et al., 2025).

## 1 Introduction

Nighttime light (NTL) data captures artificial light emissions at night, serving as a distinctive proxy for monitoring human activities that differ significantly from daytime observations. Recent studies have demonstrated the powerful capability of NTL in characterizing various aspects of dynamic urban processes and societal activities: from mapping urban extents, estimating economic vitality and energy consumption, to evaluating housing vacancy rate, disaster and conflict impact, light pollution issues (Cao et al., 2009; Chen et al., 2015; Davies et al., 2023; Elvidge et al., 1997; Li and Zhou, 2017; McCallum et al., 2022; Wang et al., 2020; Zheng et al., 2022b; Zhou et al., 2014). These applications typically use yearly or monthly temporal resolutions of NTL, such as Operational Linescan System onboard Defense Meteorological Satellite Program satellites (DMSP) (Elvidge, 1997), which are valuable for studying long-term human activity and urban development.

With advances in satellite remote sensing, higher temporal and spatial resolution NTL data have become available, enabling more detailed and timely monitoring of human activities. The Day/Night Band (DNB), a key component of the Visible Infrared Imaging Radiometer Suite (VIIRS), is carried aboard the Suomi National Polar-orbiting Partnership (S-NPP) and Joint Polar Satellite System (JPSS) satellites. This sensor captures high-resolution global nighttime data in the visible and near-infrared (NIR) spectrum, enabling daily monitoring of nocturnal light emissions. NASA's Black Marble nighttime lights product suite (VNP46) has been providing daily NTL datasets based on VIIRS DNB records since January 2012 with a spatial resolution of 500 m and a temporal resolution of one day (Román et al., 2018; Miller et al., 2012), supporting more potential applications for short-term human activity monitoring. Among the current NASA Black Marble product suites, the VNP46A2 dataset has been daily moonlight- and atmosphere-corrected based on daily at-sensor TOA nighttime radiances (VNP46A1). Despite these improvements, several challenges remain for daily NTL products, especially regarding data quality and reliability (Wang et al., 2021; NASA, 2018). Notably, spatial observational coverage mismatch (hereafter spatial mismatch, i.e., misalignment between satellite footprint and output grid) (Campagnolo et al., 2016; Román et al., 2018; Wolfe et al., 1998), angular effect (i.e., systematic variations in observed NTL radiance with changing viewing zenith angle) (Tan et al., 2022) and missing data holes (i.e., scattered pixels flagged as low-quality or missing values in cloud-free images) have all been shown to introduce significant uncertainties into daily NTL series (Campagnolo et al., 2016; Wolfe et al., 1998). These issues can lead to spurious fluctuations or gaps in the observed NTL intensity that do not reflect actual changes in human activity, thereby limiting the accuracy and reliability of short-term human activity monitoring based on daily NTL data (Fig.1; Hu et al., 2024; Wang et al., 2021).

Specifically, the spatial mismatch error arises when the 740m DNB view footprint misaligns with the 500m gridded output pixels, leading to brightness measurement errors (Hu et al., 2024) (Fig. 1a), reported as the relatively large temporal daily radiance variation observed (Román et al., 2018) (Fig. 1b). Estimating the error using any model or formula is challenging since the discrepancy is random from day to day. To mitigate the spatial mismatch error, it has been recommended that a 3-by-3-pixel averaging window be applied (Román et al., 2018). However, this approach potentially causes a "blooming effect" where the signal in specific pixels becomes diffused.

The anisotropic behavior of artificial light emissions and observation—dependent on local landscape and satellite viewing angles—is yet to be addressed in VNP46A2. The S-NPP satellite experiences daily variations in the sensor's viewing zenith angle (VZA) over a 16-day orbital cycle (Tan et al., 2022). The DNB captures both direct artificial light (e.g., street lamps, vehicle headlight, indoor light through a window) and reflected light (Fig. 1a). Due to the diversity of built environments, surrounding buildings and trees create a blocking effect on light sources. For some top-covered light sources, such as streetlights and indoor lighting, the observed radiance may change as the satellite's viewing angle varies. For example, these lights might not be observable from a nadir view but could be detected from off-nadir angles (Tan et al., 2022). Additionally, reflected light sources, in addition to being influenced by blocking effects and visible changes, are inherently anisotropic and are sensitive to surface reflectance (Barnsley et al., 1994). Collectively, these factors result in pronounced angular effects, which are directly reflected in the variability of NTL with respect to VZA (Wu and Li, 2024) (Fig. 1c). In urban areas, this variability is closely tied to the complexity of the built environment, with Tan et al. (2022) identifying negative, U-shaped, and positive angular effect across different built environments. In general, the angular effect introduces uncertainties in the NTL time series, potentially leading to inaccuracies in estimating dynamic changes in ground truth conditions (Wang et al., 2021). Therefore, using angularly consistent NTL observations is essential to ensure the reliability and accuracy of NTL-based applications.

In addition, a certain number of pixels in cloud-free images from the VNP46A2 were masked as low quality or without valid pixel values. As a result, these cloud-free images exhibit small missing holes (see an example in Fig. 1d). Unlike pixels covered by actual clouds and snow, these small missing holes are spatially or temporally discontinuous. Some small missing holes inherently lack valid NTL values in the DNB, and another portion of holes are flagged as suspect VIIRS Nighttime Cloud Mask (VCM) detections and subsequently output as poor-quality mandatory QA flags (Wang et al., 2021). The benchmark test conducted during the Black Marble production illustrates that the accuracy of the quality mask varies depending on factors such as high albedos, atmospheric and geographic conditions (Wang et al., 2020). These holes pose challenges for subsequent applications, since we observe a higher proportion of missing data holes in areas with higher annual average NTL intensity, which often correspond to city cores (Fig. 1e). The high albedos of land surfaces and the heavy air pollution in urban cores may contribute to the errors in the quality mask. Therefore, it is crucial to interpolate these small missing holes to ensure the continuity of cloud-free NTL images and enhance the practical application of NTL images in urban core areas.

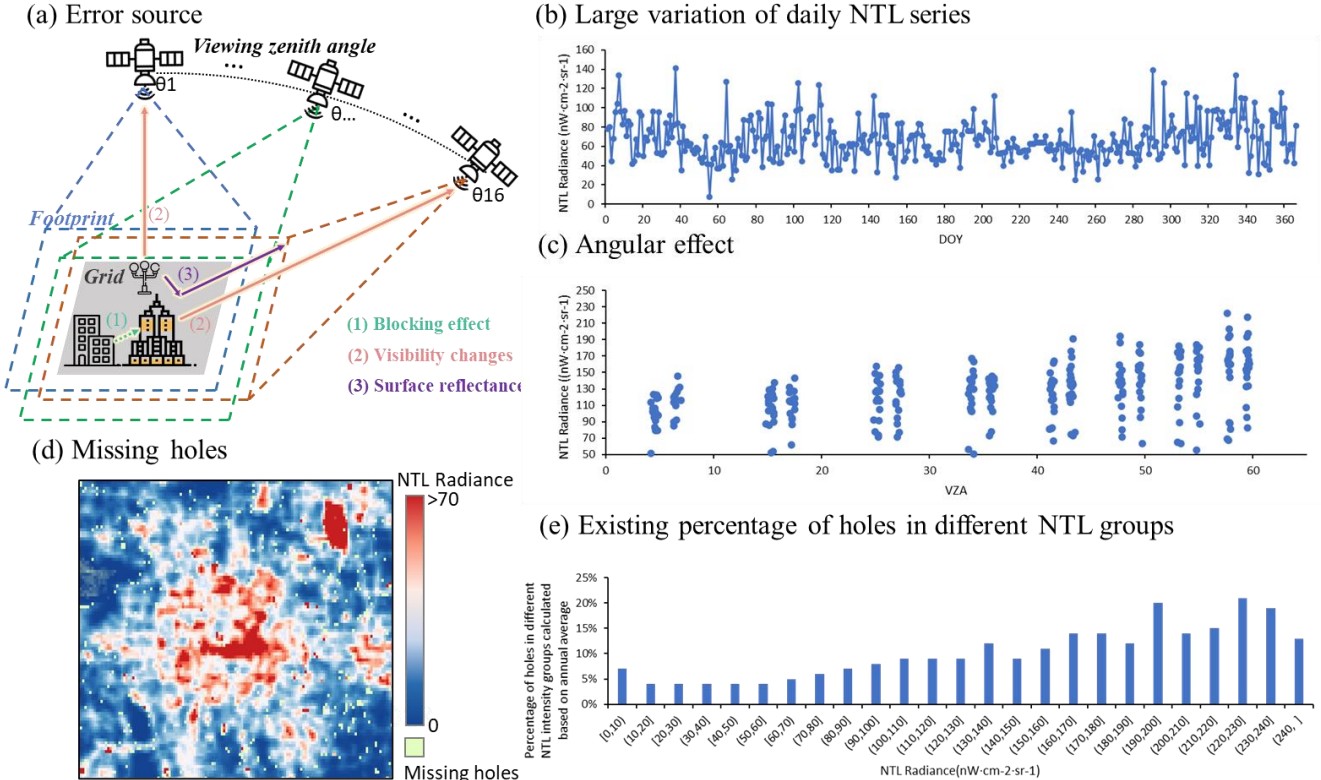

**Figure 1: An illustration of spatial mismatch error, angular effect error and missing data holes of VNP46A2. (a) Error source: Schematic diagram illustrating the existence of spatial mismatch and angular effects, (b) The instability of daily NTL series, (c) Angular effect: The changing trend of daily NTL intensity along with VZA changes within one year, (d) Small holes existence in cloud-free NTL image of Beijing on October 1, 2018, (e) The percentage of holes in different NTL intensity groups calculated based on annual averages.**

Until an effective solution is developed, researchers rely on aggregation methods, such as calculating weekly or monthly averages, to minimize the impact of uncertain errors in daily time series in regional research cases (Alahmadi et al., 2021; Zhou et al., 2022). It becomes crucial to establish a temporally consistent daily NTL dataset to conduct a quantitative analysis of human activities at finer time scales. A Self-adjusting method featuring Filter and Angular effect Correction (SFAC),

proposed by Hu et al. (2024), provides an effective solution to mitigate errors arising from spatial mismatch and angular effects. The SFAC only requires the original NTL data and does not require other auxiliary data for error correction. Therefore, it has strong generalization ability and is conducive to producing a dataset. While SFAC effectively addresses spatial mismatch and angular effects, the issue of small missing holes remains unresolved, which leads to incomplete temporal and spatial coverage, especially in urban central areas (Fig.1e). In addition, using the annual average value as the reference for correcting the angular

effect can easily mask the satellite observation angle information of the corrected data, leading to inconsistent-angle adjusted results. Owing to surface heterogeneity, the angular effect differs across locations, and corrections based on unknown observation angles may compromise the spatial comparability of the data.

This study aims to address these gaps by developing a refined dataset that enhances the accuracy and reliability of daily NTL observations. Such improvements are critical for facilitating more precise short-term urban dynamics and human activities analyses. To achieve this, we refined and optimized the SFAC method by adjusting the data to a consistent satellite monitoring angle and developed a spatiotemporal interpolation method to ensure the completeness of cloud-free images. Focusing on 653 major cities with populations projected to exceed 1,000,000 by 2025, our dataset targets densely populated urban centers worldwide. As population and economic activity hubs, these cities play a crucial role in shaping the nocturnal landscape visible through remote sensing. It is important to note that the framework developed in this study is versatile and can be applied to any region, including small cities, towns, and rural areas, to produce high-quality daily NTL time series. We will make the code from this study available for users who wish to generate such data beyond the 653 major cities. Our work aims to support sustainable urban development initiatives globally by delivering accurate and detailed daily NTL data.

## 2 Data

### 2.1 Study area

Large cities, especially those with populations exceeding one million, are increasingly recognized as key drivers of global economic, cultural, technological, and social development. These cities are at the forefront of global challenges such as climate change, energy transition and rapid urbanization. Prioritizing the growth and sustainability of these cities is paramount for fostering global prosperity and sustainable development. To aid these key areas with valuable NTL insights, and help their urban dynamics monitoring, our study focuses on producing a high-quality daily nighttime light dataset (HDNTL) for 653 cities with a projected population of more than 1,000,000 by 2025, based on the World Urbanization Prospects (WUP): The 2018 Revision (United Nations, 2018). The location of the 653 cities is shown in Fig. 2. The HDNTL dataset generation process is not limited by city size or location, as it relies on pixel-level processing of NTL data. To facilitate broader application, we will make our code open-source, enabling researchers and practitioners to generate HDNTL datasets for other cities of interest.

The extent of each city, defined as its Region of Interest (ROI), is adaptively determined based on the GHS Degree of Urbanisation Classification (GHS-DUC) dataset (Schiavina et al., 2021). Our goal is to ensure that the ROI for every city adequately covers the contiguous urban core area radiating from the city center. The process for delineating each ROI is as follows: (1) The GHS-DUC data are reclassified into three categories: urban core area ("city", "dense town", and "semi-dense town" in GHS-DUC ), suburban and rural area ("suburbs or peri-urban area", "village", "dispersed rural area", and "mostly uninhabited area" in GHS-DUC ), and non-land area (including oceans and inland water bodies, categorized as "not on land" in GHSL); (2) Starting from the central coordinates of each city, a square region is iteratively expanded until the total urban core area is smaller than the combined suburban and rural area within the square. The expansion stops once this condition is met, and the final square boundaries are recorded as the city's ROI. For cities where the initial ROI did not fully encompass the contiguous urban core area, we manually adjusted the ROI to guarantee complete coverage. For special cases involving

urban agglomerations (e.g., Los Angeles-Long Beach-Santa Ana), we also performed visual inspections and manual adjustments to ensure that the ROI covered the core built-up areas of all cities within this urban agglomeration. In regions where cities are in proximity, some ROIs may overlap, as illustrated by the close-up of cities in the Yangtze River Delta in China (see Fig. 2); no further adjustments were made in these cases.

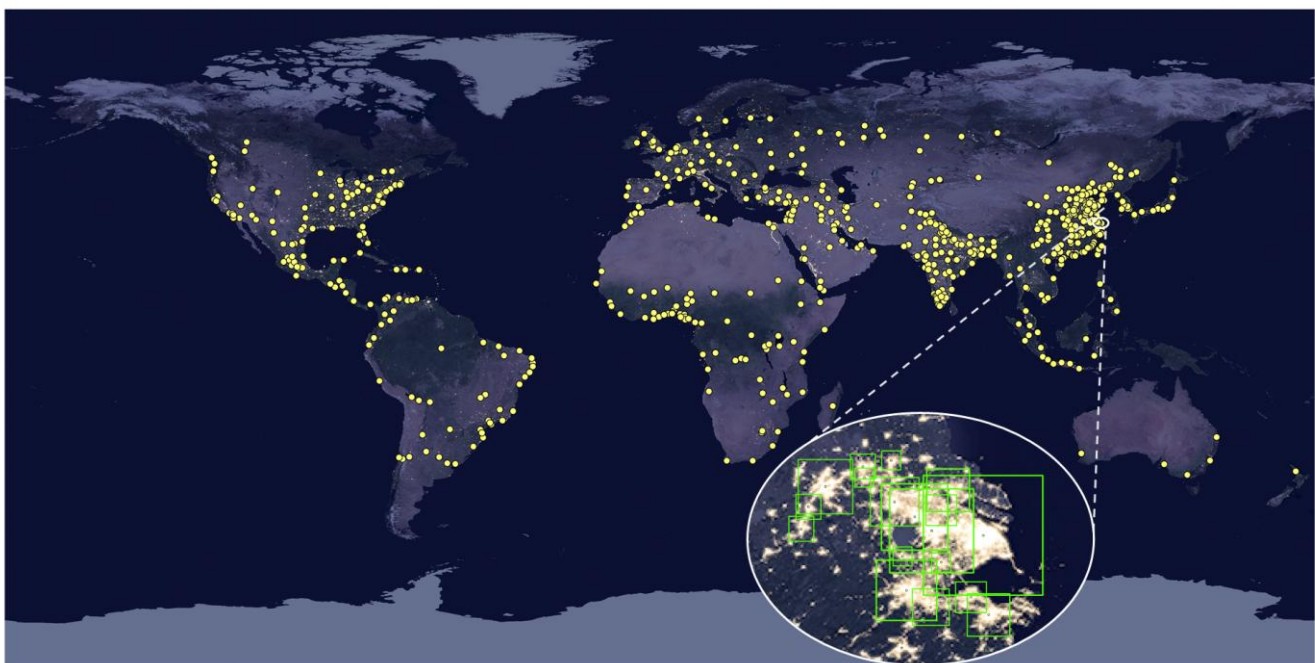

**Figure 2: Location of the 653 cities included in HDNTL and a close-up of the ROIs of cities in part of the Yangtze River Delta in China. Background NTL image from NASA (https://earthobservatory.nasa.gov/features/NightLights).**

### 2.2 Data source and preprocessing

We obtained daily VNP46A2 data from 2012 to 2024 for 653 cities, and used the "DNB_BRDF-Corrected_NTL" band as the nighttime light data to be further corrected (NASA, 2018). At the same time, the "Sensor_Zenith" band of VNP46A1 data 150 was obtained as a reference for angular effect correction (NASA, 2012). Regarding quality control, we used the mandatory quality flag, snow cover flag, and cloud mask quality flag to screen high-quality observations.

To verify the effectiveness of the data correcting process, we compared the processed data with the unprocessed data to investigate whether the spatial mismatch problem and the angular effect problem were effectively alleviated. In addition, to verify the dataset's ability to reflect the ground artificial light changes, we collected high spatial resolution NTL datasets 155 represented by SDGSAT-1 data and a power outage statistic report as comparative verification data. The SDGSAT-1 is equipped with a glimmer imager capable of capturing low-light nighttime data with a revisit cycle of 11 days. We computed the grayscale brightness from the RGB band (40m resolution) as a comparative reference. To avoid the spatial mismatch between the HDNTL and SDGSAT-1, we resampled both datasets to 1 km during verification. Given the high spatial resolution of SDGSAT-1 data, we have grounds to believe that the resampled SDGSAT-1 is less prone to the spatial mismatch issue

observed in VNP46A2 data. Due to the limited availability and quality of SDGSAT-1 data, we only selected the SDGSAT-1 NTL images of Tianjin, China, on January 25 and February 21, 2022, for comparisons (https://www.sdgsat.ac.cn/).

    NTL can directly reflect the lighting pattern of artificial lights at night, directly responding to power outages. The validity of the dataset can be evaluated by comparing the power outage response before and after data processing. On September 20, 2017, Hurricane Maria landed in Puerto Rico as a Category 4 storm, becoming the most serious hurricane to affect the island

since 1928. After the hurricane, there was a large-scale and long-term power shortage across the island. We obtained the Hurricanes Nate, Maria, Irma, and Harvey Situation Reports provided by the U.S Department of Energy (https://www.energy.gov/ceser/articles/hurricanes-nate-maria-irma-and-harvey-situation-reports), screened the daily reports 10 days before the hurricane and one month after the hurricane, and selected the percent of total customers without power in the report as the parameter for calculating the official power supply index.

## 3 Methodology

    As Fig. 3 shows, we proposed a four-step framework to generate the HDNTL dataset based on VNP46A2 after data preprocessing. During preprocessing, only the highest-quality observations are retained based on the mandatory quality flag, snow cover flag, and cloud mask quality flag. The first step was to correct the spatial mismatch, the second step was to fix the angular effect, and the third step was to interpolate the small missing holes based on spatial and temporal information. Finally,

a thorough dataset validation was conducted, including examining the effectiveness of spatial mismatch and angular effect corrections, interpolation accuracy, comparing results with high-resolution NTL datasets, and investigating the application capability in different events.

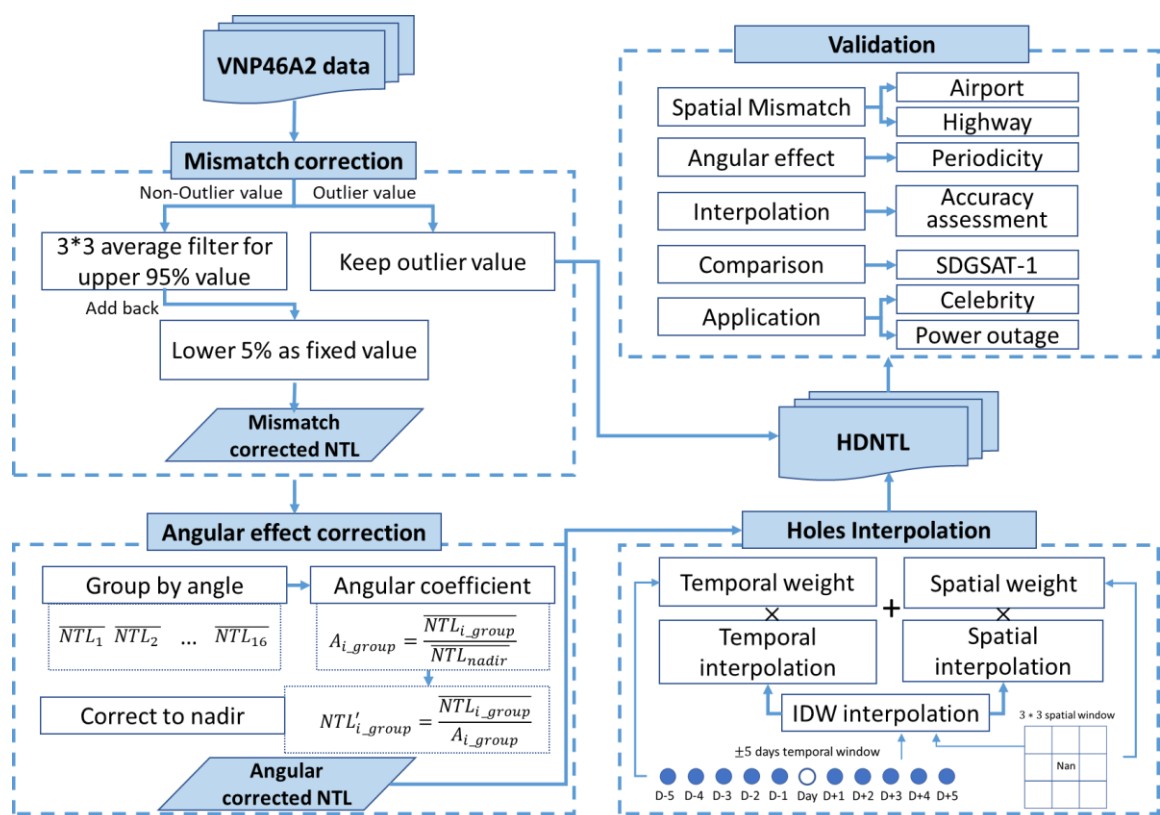

**Figure 3: Flow chart of the generation of HDNTL.**

## 3.1 Spatial mismatch and angular effect correction

We referred to SFAC (Hu et al., 2024) to eliminate the spatial mismatch and angular effect. The SFAC approach decomposes the observed radiance within a sensor's 740 m footprint into two key elements: a component originating from a fixed, consistently observed area within the scope of the sensor's footprint and remains unaffected by spatial mismatch, and a dynamic component contributed by peripheral regions due to variations in observational coverage. Based on this decomposition, the NTL in the SFAC framework is formally expressed as:

$$NTL = A * (NTL_{fix} + NTL_{variation}) + NTL_{mismatch} + \varepsilon ,\tag{1}$$

Where $A * (NTL_{fix} + NTL_{variation})$ is the light from the fixed area, $NTL_{fix}$ means the regular light emission from the fixed area, $NTL_{variation}$ represents the changes in NTL that are extremely different from regular patterns due to specific events. $NTL_{mismatch}$ is the light that exists at the areas within the daily changing footprint but outside the fixed area (Fig. 1a); $A$ is the angular effect coefficient that helps correct NTL at different observation angles to the nadir observation. $\varepsilon$ denotes residual random noise, which could originate from initial sources or be introduced during the product generation process.

The SFAC method employs a specialized A-average filter to address spatial mismatch effects. This approach operates under the principle that the fixed area (representing regular light emissions) is inherently smaller than the daily varying sensor

footprint. Accordingly, the annual lowest light intensity within a given year is used to approximate the regular light intensity coming from the fixed area ($NTL_{fix}$), except in cases of abrupt light reduction events such as power outages ($NTL_{variation}$). Therefore, the A-average filter processes identified and removed the $NTL_{variation}$ firstly using a 3 times standard deviation threshold relative to the annual mean light radiance (Hu et al., 2024; Pukelsheim, 1994). The dates with $NTL_{variation}$ observations are marked. The $NTL_{fix}$ is then determined as the mean of the lowest 5% NTL radiance for non-variation days. Subsequently, the $NTL_{mismatch} + \varepsilon$ is obtained by subtracting $NTL_{fix}$ from NTL without $NTL_{variation}$. The SFAC applied a $3 \times 3$ pixel averaging filter for $NTL_{mismatch} + \varepsilon$, and add back $NTL_{fix}$ to complete the correction process of spatial mismatch. For those days when extreme events occurred ($NTL_{variation}$), the SFAC chose to directly retain the original values in the final dataset, which not only preserved the event signal but also avoided the diffusion of the event signal component during the processing.

The anisotropic characteristics of NTL and their relationship with satellite viewing angles have been well documented and modeled (Tan et al., 2022; Li et al., 2019). The S-NPP satellite adopts a sun-synchronous polar orbit and completes a revisit cycle every 16 days. This orbital configuration ensures that each ground location maintains essentially the same VZA observation sequence within the 16-day cycle in a given year. For each pixel, the satellite observation angles throughout a year fall into 16 discrete VZA groups. The differences between groups are much larger than the variations within each group. Consequently, the annual variation in observation angles for a given pixel is not particularly complex. Moreover, pixel-based model fitting is time consuming for data generation in large areas. Therefore, we still follow the SFAC method of Hu et al. (2024), as the group-based correction approach is sufficient to address the angular effect. In the production process of the HDNTL dataset, we made subtle improvements to the angular effect correction in the SFAC method by shifting from an annual average-based correction approach to a nadir observation-based correction approach. This adjustment is necessary for several reasons. First, NTL images are frequently affected by clouds, snow, and atmospheric conditions, resulting in data gaps. Since angular effects have been well-documented, these missing data can lead to insufficient observations at certain angles. When using valid observations for the annual average calculation, observations at some angles may be overrepresented, while others may be underrepresented. This imbalance can introduce additional uncertainties into the angular effect correction process. Additionally, the annual average method may introduce regional biases, particularly in areas exhibiting distinct angular effect patterns (Tan et al., 2022). Since the annual average may represent different viewing angles in different regions, this inconsistency in angular observations can reduce the spatial comparability of the dataset. Such biases can undermine the reliability of the dataset for cross-regional analyses, such as comparing urbanization trends or energy consumption patterns between cities. By adopting a nadir observation-based correction approach, we can effectively mitigate these biases, ensuring more consistent and spatially comparable NTL data representation across diverse built environments.

Based on the periodic characteristics of VZA, we divided the daily NTL time series for each year into 16 groups according to the day sequence within the 16-day cycle. For each group, we calculated the mean NTL radiance. The group with near-nadir

observations (VZA < 6°) was designated as the reference group for angular effect correction. The angular effect coefficients and correction value of the remaining groups were calculated as follows:

$$A_{i\_group} = \frac{\overline{NTL_{i\_group}}}{\overline{NTL_{nadir}}} \tag{2}$$

$$NTL'_{i\_group} = \frac{\overline{NTL_{i\_group}}}{A_{i\_group}} \tag{3}$$

Where $\overline{NTL_{i\_group}}$ represents the average light radiance of the $i\ th$ angular group, $\overline{NTL_{nadir}}$ is the average light radiance of the reference nadir group (VZA < 6°), $A_{i\_group}$ is the angular effect coefficient for the $i\ th$ group. $NTL'_{i\_group}$ is the angular-corrected NTL radiance for $i$-th group.

To ensure the reliability of the near-nadir reference value and prevent it from being unduly uncertain by an insufficient number of observations, we established a tiered filling strategy for constructing the angular effect reference layer. Specifically, 235    the near-nadir observation group of the target year is used as the primary reference only if it contains more than three days of valid observations. Otherwise, we extend the temporal window to include near-nadir observations from both the target year and its two adjacent years (except for boundary years such as 2012 or 2024, where only one adjacent year is available), and calculate the average value of all near-nadir observations. If the combined observations still do not exceed three days, the annual mean of all observations from the target year—regardless of viewing angle—is used as the fallback reference to 240    guarantee completeness of the reference layer. The data source of each pixel in the final nadir reference image is systematically recorded in the Nadir Reference Flag, with values of 1, 2, and 3 indicating the primary (target year near-nadir), secondary (multi-year near-nadir), and tertiary (target year annual mean) references, respectively.

### 3.2 Spatiotemporal interpolation of small holes

Unlike conventional spatial or temporal-filling methods, spatiotemporal filling methods leverage more information and 245    achieve higher accuracy (Hao et al., 2023; Tan and Zhu, 2023). In this study, we developed an efficient interpolation method that integrates spatial and temporal information to estimate pixel values of small holes. Importantly, we do not interpolate missing pixels caused by large and long-persistent clouds as such interpolation is considered unreliable (Román et al., 2018). Our method is specifically designed to fill only small, spatially discontinuous gaps in otherwise high-quality data, where interpolation is more likely to be reasonable. To distinguish small missing holes from those caused by extensive coverage of 250    the cloud mask, we define the missing value hole as a missing value pixel with valid neighboring observations in the spatial windows. Considering the significant heterogeneity of urban areas, we chose the smallest spatial unit for spatial interpolation, a 3×3 window. To avoid overfilling edge pixels near coasts, lakes, or ROI boundaries, we only filled missing-value pixels that had at least four valid reference pixels within the 3×3 window. As for temporal interpolation, it has been proven that for VNP46A2 data, an average of 16 days of compositing is required to ensure at least 95% effective pixel coverage of the city 255    (Zheng et al., 2022a). Therefore, we tested the single-sided temporal window from 1 to 8 days. The temporal interpolation

window was finally determined based on the evaluation of interpolation accuracy and effectiveness. The interpolation process is as follows:

$$NTL_{is} = \sum_{n}^{N} \omega_n NTL_n \tag{4}$$

$$NTL_{it} = \sum_{m}^{M} \omega_m NTL_m \tag{5}$$

$$W_{is} = \frac{\sum_{n}^{N} \omega_n \cdot valid_n}{\sum_{n}^{N} \omega_n \cdot 1} \tag{6}$$

$$W_{it} = \frac{\sum_{m}^{M} \omega_m \cdot valid_m}{\sum_{m}^{M} \omega_m \cdot 1} \tag{7}$$

$$valid_n = \begin{cases} 1, & NTL_n \ is \ valid \\ 0, & NTL_n \ is \ invalid \end{cases} \tag{8}$$

$$NTL_i = a_{is} NTL_{is} + a_{it} NTL_{it} \tag{9}$$

$$a_{is} = \frac{W_{is}}{W_{is} + W_{it}}, a_{it} = \frac{W_{it}}{W_{is} + W_{it}} \tag{10}$$

Where $NTL_{is}$ is the spatial fill value for pixel $i$, $n$ is the $n$-th pixel in a 3×3 window centered on pixel $i$, $\omega_n$ is the inverse distance spatial weight of pixel $n$, $NTL_n$ is the NTL of pixel $n$; $NTL_{it}$ is the temporal fill value for pixel $i$, $m$ is the $m$-th day in the temporal window centered on the target date for pixel $i$, $\omega_m$ is the inverse distance temporal weight of pixel $m$, $NTL_m$ is the NTL of pixel $i$ at day $m$; $W_{is}$ and $W_{it}$ represent the valid weight ratios in the spatial and temporal interpolation processes, respectively. $NTL_i$ is the spatial-temporal interpolated NTL for pixel $i$, $a_{is}$ is the relatively spatial weight, $a_{it}$ is the relatively temporal weight. The above interpolation process was conducted for data after spatial mismatch and angular effect corrections and only applied to missing pixels with at least four valid neighboring observations in the spatial windows. $NTL_{variation}$ is also excluded during interpolation and added back after interpolation.

## 3.3 Validation strategy

### 3.3.1 Spatial mismatch and angular effect correction

Spatial mismatch errors can cause large fluctuations in the time series of NTL data. Therefore, the effectiveness of spatial mismatch correction can be evaluated by investigating the daily NTL data of regions with relatively stable lights. In addition, to eliminate the superposition effect of angular effect correction, we selected airports and highways without surrounding obstructions as sample sites to conduct this evaluation for the following reasons: (1) As functional areas with transportation attributes, airports and highways generally do not experience large fluctuations in daily light changes, are less affected by extreme events, and tend to be more stable in time series. (2) Airports are generally built in open areas, and the highways we selected generally have no surrounding buildings or other construction land obstructing their view. So both of them are less affected by angular effects. To validate the effectiveness of spatial mismatch correction, we calculated the normalized standard deviation both spatially and temporally within the case study area. Furthermore, in the highway case, due to the uniform distribution of streetlights along these highways, the corrected NTL values of highway pixels are expected to exhibit greater

spatial consistency. We therefore quantified the spatial variability (SV) of all highway pixels using the normalized annual mean and normalized standard deviation, calculated as:

$$SV = \frac{std(N)}{mean(N)} \tag{11}$$

The SV helps quantify inter-pixel contrast, with lower values indicating lower radiance differences between pixels.

Since periodic VZA induces observed periodicity of daily NTL series (Tan et al., 2022), the periodic weakening of the NTL with VZA changes can prove the effective correction of the angular effect. To assess the periodicity, we employed the periodic basis vectors following the approach of Liu et al. (2019). Given the 16-day periodicity of VZA, 16 orthogonal vectors $ei$ ($i = 1, \ldots, 16$) were established, each representing distinct VZA variations. The time series data were projected onto these orthogonal vectors using principal component analysis. The periodicity for each day within the cycle ($i = 1, \ldots, 16$) was quantified by computing the ratio between the projection vector's norm and the NTL radiance time series vector's norm, expressed as the Information Rate (IR) (Jia et al., 2023). The higher the IR, the higher the periodicity of the 16-day cycle. The overall periodicity ($i = 1, \ldots, 16$) was determined by aggregating all individual periodicities:

$$IR_i = \sum_{j=1}^{n} IR_i^j \tag{12}$$

Where $IR_i$ is the $IR$ of a pixel $i$. $IR_i^j$ is the $IR$ of pixel $i$ of $j$ $th$ day in the cycle. $n$ is the length of the cycle. To gain a regional evaluation, we calculated the average value of $IR_i$ of all pixels (total number $= N$) inside a given region as an assessment metric, the equation is:

$$IR^{region} = \frac{1}{N} \sum_{i=1}^{N} IR_i \tag{13}$$

### 3.3.2 Spatiotemporal interpolation of small holes

To evaluate the accuracy and reliability of the spatiotemporal interpolation method and determine the optimal temporal window size, we designed a comprehensive validation approach that considers both interpolation accuracy and the availability of valid reference data within the interpolation window. We selected 300 images randomly from the dataset that had already undergone spatial mismatch and angular effect corrections. In each image, we randomly masked out a subset of high-quality pixels with at least one valid value in a 3×3 spatial window, treating them as "small missing holes". The original values of these masked pixels were retained as reference data for comparison. Single-sided temporal windows were tested from 1 to 8 days to evaluate the impact of window size on interpolation accuracy, and the interpolated values were compared with the reference values using R², RMSE, and MAE as evaluation metrics. Additionally, we assessed the relative temporal weight for each window size, with higher weights indicating a greater proportion of valid reference data and higher reliability.

### 3.3.3 Short-period event detection

The corrected NTL time series should have eliminated false fluctuations to improve the monitoring sensitivity of specific events, especially the response to sudden weakening or strengthening. We selected two cases for analysis. The first case refers

to the 2023 Liuyang Fireworks Festival (LFC) held in Liuyang City, Hunan Province, China, on the evening of November 4, 2023. After a four-year hiatus due to COVID-19, the festival scale was grand, with tens of thousands of fireworks set off at 49 points in the main urban area of Liuyang. The fireworks festival also presented a variety of themed activities and night markets, attracting a large number of citizens and tourists. The event lasted until the early morning of November 5. During this period, the light intensity in this area should be significantly enhanced compared with regular dates. The second case is the Beirut Port

explosion on August 4, 2020, resulting in a death toll of at least 220 people, with over 7,000 injured and 60 people reported missing. In the affected area centered on the explosion site, a large area of buildings was damaged, and power outages occurred, leaving 300,000 people homeless. During this period, the area should have lower light intensity than usual. For the above two cases, we used the detectability index (DA) to detect whether the NTL value on the day of the event was consistent with expectations (Hu et al., 2024). We compared the detection advantage of HDNTL over the VNP46A2 dataset. When the absolute

value of DA is greater than 3 (usually used as the threshold for mutation signal detection), it suggests that the value of this event deviates statistically significantly from the normal distribution and may be an outlier or extreme value (Pukelsheim, 1994). The DA is calculated as follows:

$$DA = \frac{NTL_{event} - NTL_{mean}}{NTL_{StdDev}} \tag{14}$$

Where $NTL_{event}$ is the NTL intensity for a pixel on the event day, $NTL_{mean}$ represents the average NTL intensity for a pixel

over a given year, $NTL_{StdDev}$ denotes the standard deviation of NTL intensity for a pixel over a given year.

### 3.3.4 Estimation of power outage and restoration

In addition to assessing the ability to monitor short-term events, we also focus on whether the corrected NTL data can more accurately reflect the actual situation on the ground. Artificial light creates the NTL landscape, and power outages are directly related to the reduction of NTL. Thus, the NTL data's ability to reflect the actual situation on the ground can be evaluated by

comparing NTL changes and the statistics of power outages. We chose the power outage event in Puerto Rico, which was affected by Hurricane Maria. We collected daily power reports 10 days before and 1 month after the hurricane. The power supply rate is calculated as $1 - the\ percentage\ of\ affected\ customers$ in the report. To enhance the comparability of NTL data and power data reports, we designed a Customer-based Power Supply Index (CPSI) based on NTL data as below:

$$CPSI_t = \frac{NTL_t}{NTL_{pre-outage}} \times PD \tag{15}$$

Where the $NTL_t$ represents the NTL radiance at time $t$, $NTL_{pre-outage}$ is the mean NTL value before the outage event in one week. $PD$ is the percentage of the pixel-level population in the region-level population calculated based on the GHS-POP R2015A dataset (European Commission, Joint Research Centre (JRC), 2015). This index is calculated on the pixel level, and the total sum of $CPSI$ of each pixel will be the estimation of the outage situation for the whole region. It should be noted that, in order to conduct a holistic assessment of Puerto Rico, we used the HDNTL data production framework to generate data for

the entire area of Puerto Rico, which is inconsistent with the ROI range in the current HDNTL dataset.

## 4 Results

### 4.1 Effectiveness of spatial mismatch correction

We used two landscapes, airports and highways, with less angular effects, to demonstrate the effectiveness of spatial mismatch correction. Figure 4 shows three airport cases. The role of spatial mismatch correction is to reduce data fluctuations resulting from capturing the light sources of adjacent pixels due to different daily footprints. We used the normalized standard deviation (n-std) to see the annual fluctuation for each pixel in the case area. The first case is Beijing International Airport in 2023. A comparison of the n-std spatial distributions between VNP46A2 and HDNTL indicates that HDNTL consistently demonstrates reduced pixel-level variability in the airport region. We selected one terminal building pixel to see the annual time series, and the n-std of the NTL time series was reduced from 0.2 to 0.18, while the average did not change much. The second case is at the Los Angeles International Airport in 2023. The n-std distribution image also showed a large reduction after correction. The fluctuation of the annual time series of the selected pixel (airport runway) decreased, and the n-std reduced from 0.27 to 0.22. We can also see that the average NTL radiance of the chosen pixel increased from 38 nW·cm-2·sr-1 to 54 nW·cm-2·sr-1, indicating that the NTL is significantly underestimated due to spatial mismatch, as there are no obvious lights outside the runway. The third case is Sydney Airport in 2023. We also chose the runway pixel. The runway of Sydney Airport is built on the sea, so it should have lower lights. However, the VNP46A2 was overestimated due to the spatial mismatch. After correction, the NTL decreased, and n-std decreased from 0.23 to 0.17.

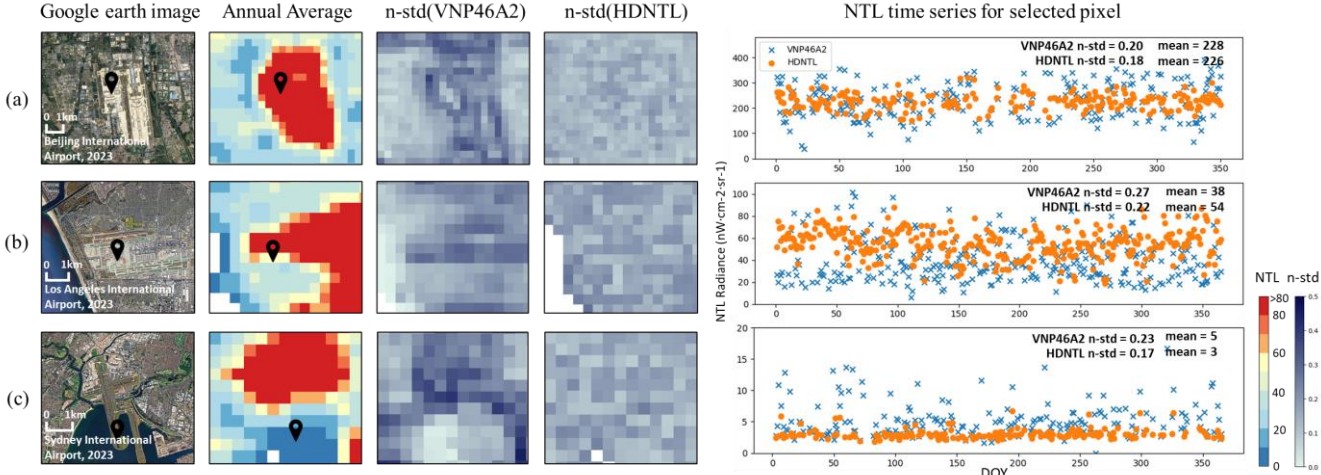

**Figure 4: Spatial mismatch correction effectiveness in less-angular effect areas within airport cases. (a) Beijing International Airport, (b) Los Angeles International Airport, (c) Sydney International Airport. Google Earth images © 2025 Google LLC.**

A highway without surrounding obstructions that affect satellite observations can serve as another good sample to validate the effectiveness of spatial mismatch correction. The chosen sections—Tianfu Avenue South Section 2 in Chengdu and the Beijing–Hong Kong–Macao Expressway in Zhengzhou—have no surrounding buildings or other construction land obstructing their view, meaning their lighting mainly comes from streetlights and vehicle headlights, and less angular effect (Chang et al.,

2019) (Fig.5). Given that the spatial resolution of the NTL data is 500 meters, the pixel width is sufficient to cover the highway

width, yet adjacent pixels remain highly susceptible to spatial mismatch effects. A transect analysis across each highway was conducted to evaluate the spatial mismatch correction response (Fig. 5). In both the Chengdu and Zhengzhou cases, correction resulted in a more pronounced contrast in radiance intensity between the highway pixels and the non-urban land on both sides, manifested as a narrowing of the peak in the NTL intensity profile. Moreover, due to the uniform distribution of streetlights along these highways, the corrected NTL values of highway pixels are expected to exhibit greater spatial consistency. We

therefore quantified the SV of all highway pixels. We observed a decrease in SV following the correction in Fig.5. This result confirms the effectiveness of the spatial mismatch correction.

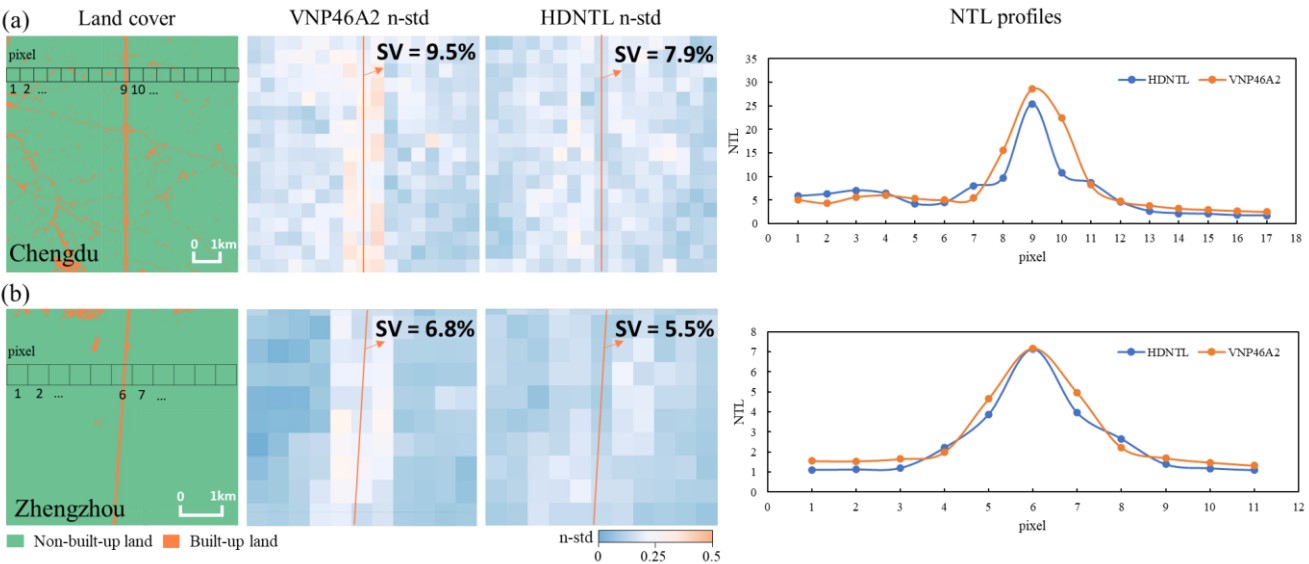

**Figure 5: Spatial mismatch correction effectiveness in less-angular effect areas within highway cases. (a) Chengdu Tianfu Avenue, (b) Zhengzhou section of Beijing-Hong Kong-Macao Expressway. Land cover data from the European Space Agency (ESA)**
**WorldCover 10 m 2020 product © ESA WorldCover Consortium.**

## 4.2 Effectiveness of angular effect correction

Previous literature has proved that the angular effect is highly correlated with the surface built-up landscape (Tan et al., 2022, 2023). When VZA changes, the blocking effect and visibility changes may lead to the angular effects in different directions. Specifically, areas with dense and high buildings, such as urban core areas, tend to have a negative angular effect,

that is, NTL decreases with increasing VZA. A U-shaped angular effect tends to be produced in the transition area between urban core areas and suburban bungalows. Suburban areas where bungalows are the main buildings tend to have a positive angular effect because the blocking effect is reduced. We selected three representative areas from 653 cities with different surface built-up landscapes to explore whether the IR related to the angular effect was significantly reduced after data correction.

For the first case, we selected part of the core area of Tokyo. The Google Earth 3D map (Fig. 6a) shows many high-rise buildings here with a high density. We drew the NTL series of one pixel of this region in 2021. Due to the blocking effect of high-rise buildings, the changing trend of NTL with the increase of VZA was consistent with the negative angular effect. After correction, this trend was no longer apparent, and the IR showed that the periodicity caused by the angular effect after treatment was reduced from 0.390 to 0.138, with a reduction rate of 64.62%. The second case is a residential and industrial area near the

core area of Sao Paulo, Brazil, with a relatively complex building structure. From the NTL time series of one pixel, the area here had an apparent U-shaped angular effect, that is, with the increase of VZA, NTL first decreased and then increased. After correction, this trend was also significantly weakened, and the periodicity of IR was reduced from 0.398 to 0.141, with a reduction rate of 64.57%. The third scenario pertains to Toronto, Canada, focusing on a residential locality far from the city center. The buildings here predominantly feature flat and low-rise buildings. It can be seen that the NTL series before treatment

had a significant increase trend with the increase of VZA, which was a positive angular effect. After correction, this trend was also weakened, and IR was reduced from 0.382 to 0.129, with a reduction rate of 66.77%. In general, HDNTL can perform effective angular effect correction for different urban built environments, greatly enhancing the comparability of data in different spaces and times.

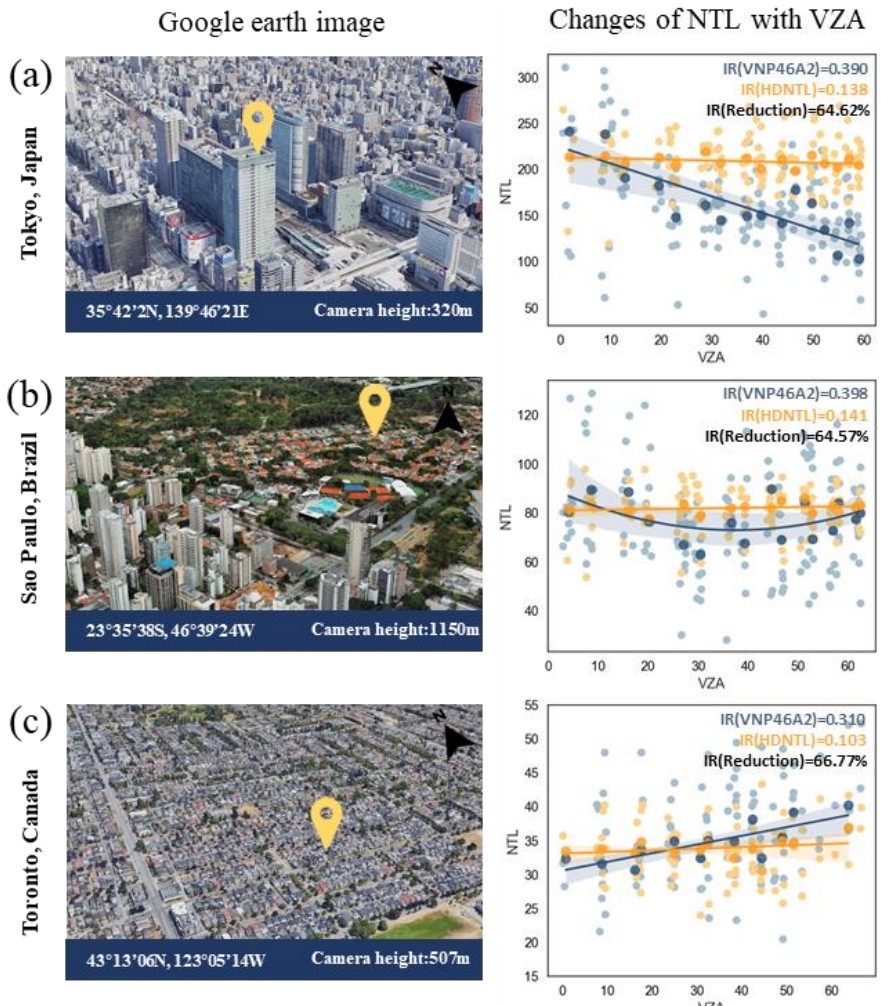

**Figure 6: Angular effect and its correction effectiveness in different built-up environments. Google Earth images © 2025 Google LLC.**

### 4.3 Effectiveness of small holes interpolation

The urban surface environment presents highly heterogeneous characteristics due to the transformation and intervention of human activities. Therefore, we chose the smallest spatial window, a 3×3 window, to perform spatial interpolation of NTL,
which can ensure the reliability of the interpolation results. In terms of temporal interpolation, we started with a single-sided window of 1 and gradually tested and verified the interpolation accuracy until 8. According to Fig. 7a, when the single-sided window was set to 5, the RMSE and MAE of the interpolation were lowest. Further, according to Fig. 7b, when the single-sided window was 5, the relative time weight was higher, which means more valid reference values can be obtained in the time dimension. After the single-sided window was greater than 5, the marginal effect of increasing the relative time weight was
weakened. Therefore, we finally chose a single-sided window of 5 for temporal interpolation. Taking into account the effective

value information of time and space, we determined the relative weights of the spatial interpolation results and the temporal interpolation results, and then performed comprehensive interpolation (Eq. (9)&(10)). We randomly masked out some valid pixels for evaluating the interpolation method, i.e., interpolating these masked-out pixels, and compared the results with their original values (i.e., reference values). As shown in Fig. 7c, the linear regression $R^2$ between the reference value and the interpolation value was 0.98, the Pearson correlation coefficient (r) was 0.99 (p< 0.001), the RMSE was 2.64, and the MAE was 1.24, indicating the high accuracy and reliability of the comprehensive interpolation method. After the hole interpolation implementation, all cities' valid pixels increased by about 2%.

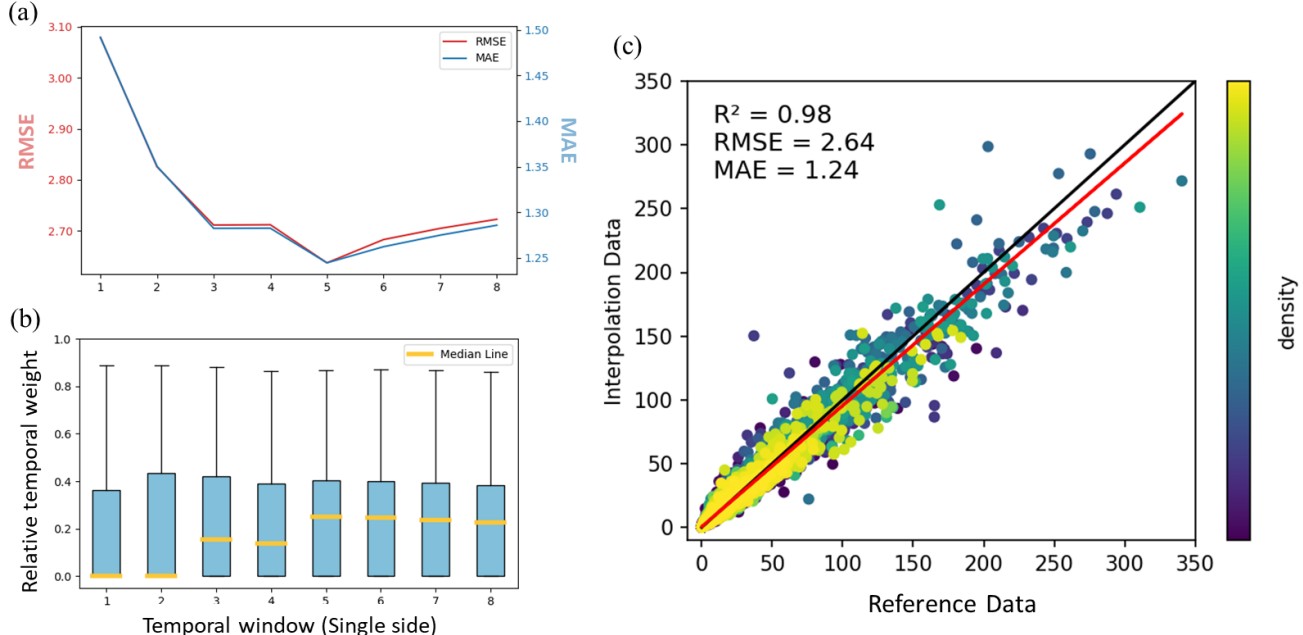

**Figure 7: Interpolation accuracy test. (a) The RMSE and MAE of interpolation by different time windows, (b) The relative temporal weight of different time windows, (c) The comparison between reference data and its interpolation by the 5-day temporal window.**

## 5 Discussion

### 5.1 Comparison with high spatial resolution NTL data regarding temporal changes

We selected high-quality and clear nighttime light observation images from SDGSAT-1 of Tianjin on January 25, 2022, and February 21, 2022. We compared their temporal changes on pixel-level with those retrieved from HDNTL and VNP46A2. Since SDGSAT-1 is high spatial resolution data, it can be concluded that after upscaling, SDGSAT-1 is less affected by mismatch issues. Thus, the change between two SDGSAT-1 images can more accurately capture the ground light changes. The advantage of the HDNTL dataset is that it restores the daily variation characteristics of NTL, so it is expected that changes retrieved from HDNTL should be closer to SDGSAT-1. As shown in Fig. 8, the box plots of the change rate between the two days of HDNTL and SDGSAT-1 are closer, showing a consistent change trend. The VNP46A2 data show a large fluctuation

in the change rate. This phenomenon reflects the significant fluctuation of VNP46A2 on the daily scale, making it challenging to capture the true characteristics of daily changes accurately. Therefore, HDNTL data appear more stable and reliable in representing NTL daily changes.

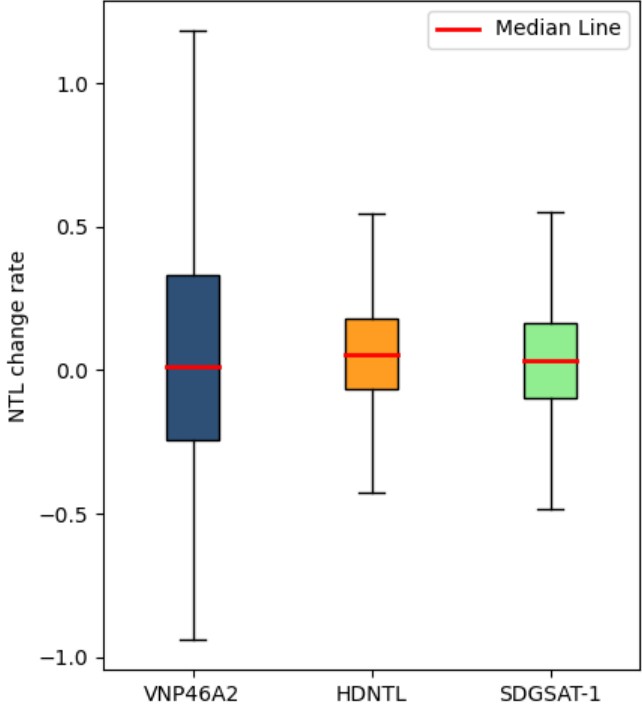

**Figure 8: Comparison of temporal changing rate between SDGSAT-1, VNP46A2, and HDNTL.**

**5.2 Event detection regarding short-period changes**

Figure 9 shows the daily NTL series of selected pixels of two short-term events in their targeted year. Figure 9a shows the daily time series of NTL for one pixel at the 2023 Liuyang Fireworks Festival (LFC) site. On the event day, the DA value calculated based on VNP46A2 was 2.1, and the DA value calculated based on HDNTL was 4.12. The absolute DA value from HDNTL was greater than 3, exceeding the mutation signal recognition threshold. For the case of a sudden drop in light, as

shown in Fig. 9b, the figure shows the daily time series of NTL for one pixel at the explosion center. Before processing, the DA on the event day was -1.76, which failed to become a mutation signal. After processing, the DA was -3.63, and its absolute value was greater than 3, which can be identified as a mutation signal. This proves that after processing with spatial mismatch correction and angular effect correction, the HDNTL dataset can reduce the fluctuation of non-real situations compared to the VNP46A2 dataset, and highlight the changes in light density caused by the mutation event.

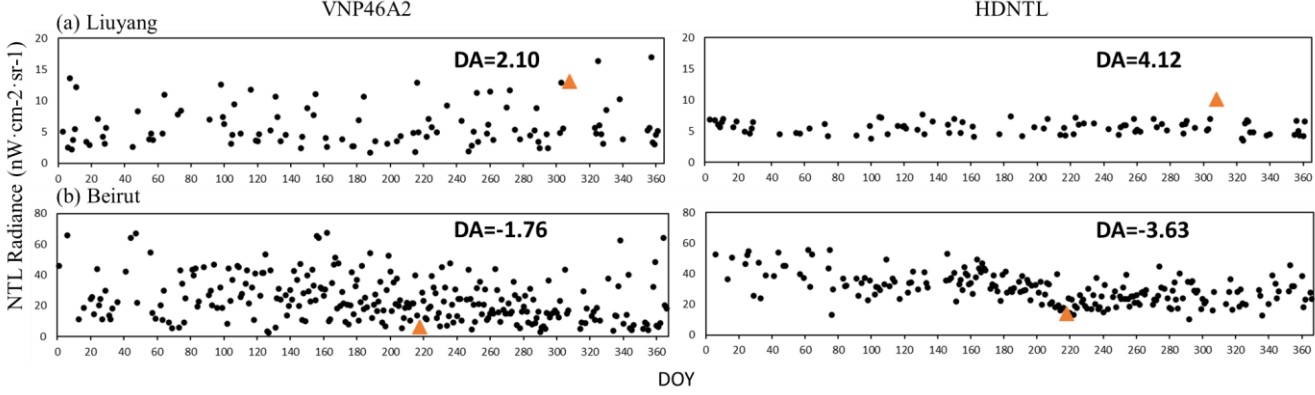

**Figure 9: Two cases for short-period event detection by NTL.**

### 5.3 Comparison with an official report regarding the power outage

According to Eq. (15), we evaluated the CPSI as the power supply index based on NTL and population density data. In this process, we chose the "Gap_Filled_DNB_BRDF_Corrected_NTL" band in the VNP46A2 data to obtain more valid values for

comparison. Figure 10 shows the results of the power supply assessment using HDNTL and VNP46A2, respectively. Also, it presents the regression $R^2$ between the time series of the two assessment results and the official report data. The results show that the assessment results of HDNTL were more consistent with the official report, with an $R^2$ value of up to 0.839, which was 170% greater than the VNP46A2 data. This result emphasizes the advantages of HDNTL in capturing changes in electricity supply and further verifies the application potential of HDNTL in urban energy and disaster impact research.

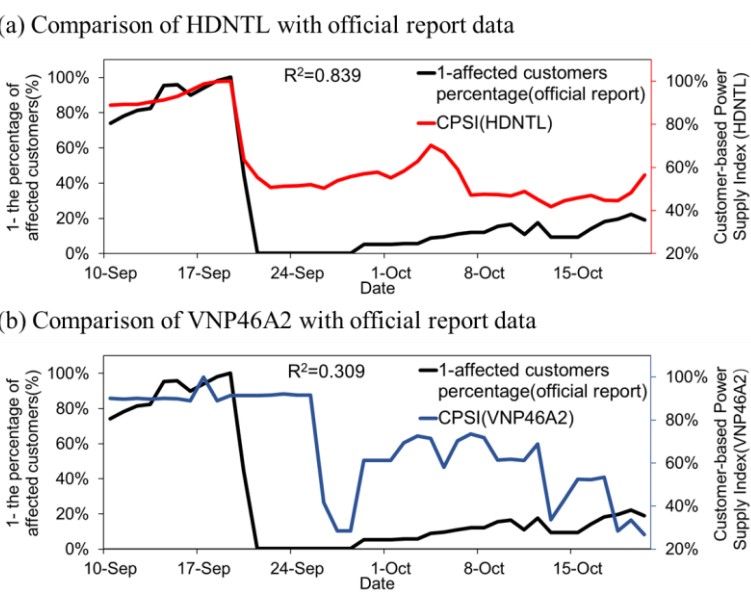

**Figure 10: Comparison of CPSI calculated based on VNP46A2 and HDNTL with official report data.**

## 5.4 Limitations

We have further improved the angular effect correction in the SFAC algorithm. In the original algorithm, the angular effect correction was based on the annual average NTL value, which would lead to some uncertainty. That is, this average value may correspond to any VZA. Existing research and the cases we discussed show that the angular effect highly correlates with the urban built environment landscape. Therefore, NTL data corrected to the same observation angle facilitate comparability between different study areas. By correcting all the data to nadir observation, theoretically, in areas with high building density and height, the NTL observation value of the nadir angle is relatively high compared with other angles. Conversely, the nadir NTL observation value is relatively low in areas with low buildings compared with other angles. Figure 11 illustrates a case in Los Angeles. Figure 11a and Fig. 11b show the corrected results based on the annual average (SFAC) and the corrected results based on nadir observation (HDNTL), respectively, both depicting the average NTL images for Los Angeles in 2024. Figure 11c displays the difference between HDNTL and SFAC. The results reveal that HDNTL data exhibits higher corrected NTL values in urban centers, particularly in areas with high-rise buildings, such as downtown Los Angeles shown in Fig. 11d. In contrast, in areas with low-rise and sparsely distributed buildings, such as the city of industry in Los Angeles shown in Fig. 11e, which features warehouses, retail stores, and car dealerships, the HDNTL data shows lower corrected NTL values. As a result, this adjustment could accentuate differences between a city's core and periphery. The HDNTL may not fully capture the facade lighting of the building, especially in low-rise building areas. Users can customize the correction parameters of the angular effect through our open-source code according to the purpose of the research.

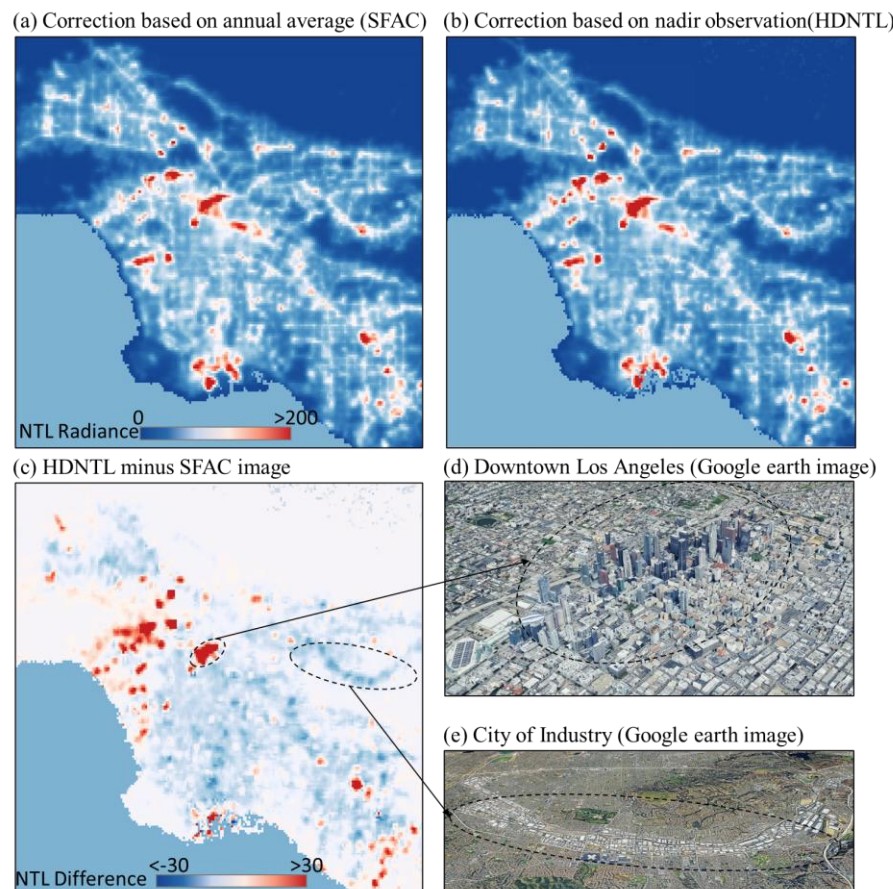

**Figure 11. Comparison of angular correction results based on nadir correction (HDNTL) and annual mean correction (SFAC). Google Earth image © 2025 Google LLC.**

## 6 Data availability

The HDNTL dataset for 653 cities from 2012 to 2024, with a spatial resolution of 15 arc-seconds (approximately 500 meters), is freely available at https://doi.org/10.5281/zenodo.17079409 (Pei et al., 2025). The data is organized in descending order of city population, with every 10 cities stored in a zip archive. City order numbers can be referenced in the data table provided in the .xlsx format. After extraction, each city's data is stored in a separate folder, containing 13 annual NTL data in GeoTIFF format. Daily data is stored as bands within each annual NTL image. To prevent confusion among end users, we removed images heavily obscured by clouds (percentage of invalid pixels >50%, pixels with constant invalid values, such as water bodies, are not considered in the calculation) from the daily NTL time series. Yearly count of daily data records available for each city in the HDNTL dataset is recorded in a .csv file for users to query. Additionally, we provide a flag layer indicating the source of the angular effect reference value and whether interpolation was applied. This flag is encoded as a two-digit integer. The tens digit (values 0, 1, 2, or 3) denotes the source of the nadir-group reference value: 0 for an invalid value, 1 for

the mean value of nadir observations from the current year, 2 for the mean value of nadir observations from the current year and its two adjacent years (except for boundary years such as 2012 or 2024, where only one adjacent year is available), and 3 for the annual mean value of all valid observations from the current year. The units digit (values 0 or 1) indicates the interpolation status: 0 for no interpolation and 1 for an interpolated value. The storage structure of this flag is consistent with that of the HDNTL data product.

## 7 Conclusion

A high-quality daily nighttime light (HDNTL) dataset for the global 653 cities (2012-2024) was produced in this study. From correction to validation, this study includes four parts: the correction of spatial mismatch, the correction of angular effect, the interpolation for small missing holes, and the validation and application. Compared to VNP46A2, the two sample areas, airport and highway, showed a good effect on spatial mismatch correction, with a decreased n-std in annual NTL time series and enhanced spatial consistency among pixels with homogeneous light sources. The angular effect correction works well on three different urban building landscapes, weakening the angular effect in different directions and decreasing the angular effect's periodicity. The spatiotemporal interpolation accuracy also showed promising results in the 5+5 temporal window and 3×3 spatial window. The spatiotemporal interpolation of missing data holes is highly similar to that of reference data, as indicated by an $R^2$ of 0.98. After the hole interpolation implementation, all cities' valid pixels increased by about 2%. Finally, the HDNTL demonstrated better consistency with SDGSAT-1 data in terms of NTL change rate and superior performance in short-period event detection and alignment with power outage report data compared to VNP46A2. Generally, our HDNTL dataset can effectively decrease the instability of the daily NTL series caused by spatial and temporal errors in the VNP46A2, enhance the comparability of the data over different time and space, and improve the ability and accuracy of the NTL to reflect the actual events on the ground. Leveraging the degree of urbanization metrics, we have defined Regions of Interest (ROIs) for 653 cities globally. The HDNTL dataset from 2012 to 2024 has been provided within these delineated areas. This dataset offers a valuable resource for enhanced analysis and assessment of nighttime human activity dynamics on a daily scale during the progression of urbanization.

## Author contributions

ZP designed and developed the methodology, performed analysis and validation, produced the dataset, and wrote the paper. XZ supported and designed the study and wrote the paper. YH designed and developed the methodology and reviewed the paper. JC developed the methodology and reviewed the paper. XT developed the validation and reviewed the paper.

## Competing interests

The authors declare that they have no conflict of interest.

## Acknowledgments

This study was supported by the Shenzhen Park of Hetao Shenzhen-Hong Kong Science and Technology Innovation Cooperation Zone "Theories for Spatiotemporal Intelligence and Reliable Data Analysis" (Project ID: HZQSWS-KCCYB-2024058), the Research Grants Council of Hong Kong (project No.15229222), the National Natural Science Foundation of China (No. 42401474), and the Hong Kong Polytechnic University (project No. Q-CDBP).

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
