# Peer review of "A high-quality daily nighttime light (HDNTL) dataset for global 600+cities (2012-2024)"

_Earth System Science Data, 2025_

## Author Comment (AC1)

**Response to Reviewers' Comments**

We appreciate comments from the editor and four reviewers on our manuscript "A high-quality daily nighttime light (HDNTL) dataset for global 600+ cities (2012–2024)", submitted to Earth System Science Data. The reviewers raise excellent points. We have read the comments carefully and made modifications. The changes in the revised manuscript were highlighted in blue.

**Reviewer #1:** This study proposed a method to correct the angular effects, spatial mismatch, and small holes in daily NTL data and generated a wonderful product of high-quality daily NTL data covering more than 650 cities across the world. This a very valuable dataset, and the generation procedures and validation assessments are reasonable and solid. I think it should be published after a revision.

**Response:** *Thanks for your constructive advice. We have carefully read your comments and thoroughly responded to your concerns. A detailed point-by-point response is listed below.*

**Q1:** In Line 97-98, the authors mentioned that the knowledge of the local building environment is important for identifying the sources of light. I am not so sure why the authors paid attention to this? It looks like there are no associations with the HDNTL generation.

**Response:** *Thank you for your valuable comment. The intention behind Lines 97-98 was to highlight a limitation of the original SFAC method in correcting for angular effects. Specifically, the original SFAC method uses the annual average value as the reference for angular correction. However, using the annual mean can obscure the actual observation angles present in the corrected data.*

*Ideally, each VZA (View Zenith Angle) group should contain 22–23 days of observations per year. However, due to missing data, the actual number of valid observations in each group is often less than 22 or 23 days. As a result, simply using the annual average value as a correction reference can mask the actual observation angle of the corrected data.*

*For example, as shown in Figure R1, we selected two pixels from the central urban areas of Guangzhou and New York City. Both exhibit a negative angular effect, meaning that NTL values decrease as VZA increases. We calculated the number of valid observations in each VZA group for 2024 and then computed the weighted average VZA for each pixel. In Guangzhou, more valid data were available at VZA > 30°, resulting in a weighted average of 39°, while in New York, the weighted average was 30°. According to the negative angular effect, since the corrected NTL for New York corresponds to a VZA of 39° and for Guangzhou to 30°, the difference in observation angles is not fully corrected, which reduces the comparability of the data between the two cities.*

*Therefore, our work improves upon the original SFAC method by calibrating all angular effects to a consistent reference—the nadir—thus enhancing the spatial comparability of the data.*

*To avoid misunderstanding, we have revised the original text as follows:*

"In addition, using the annual average value as the reference for correcting the angular effect can easily mask the satellite observation angle information of the corrected data, leading to inconsistent-angle adjusted results. Owing to surface heterogeneity, the angular effect differs across locations, and corrections based on unknown observation angles may compromise the spatial comparability of the data."

[Figure]

Figure R1: Two cases of the valid days in 16 VZA groups and the weighted average VZA. Google Earth images © 2025 Google LLC.

**Q2:** In Line 174, what does the "fixed area" mean? An area with stable light emissions?

**Response:** *Thanks for your valuable feedback. Regarding lines 171-172 in the original version, we initially stated: "This fixed area corresponds to the pixel's regular light signature, with the inter-daily brightness variations caused by footprint displacements being identified as the mismatch effect." Upon your question, we recognize this wording is too brief and could be potentially ambiguous. To better align with Hu et al. (2024)'s formal definition of the fixed area, we have revised this statement in the new manuscript to describe the fixed area more precisely:*

"The SFAC approach decomposes the observed radiance within a sensor's 740 m footprint into two key elements: a component originating from a fixed, consistently observed area within the scope of the sensor's footprint and remains unaffected by spatial mismatch, and a dynamic component contributed by peripheral regions due to variations in observational coverage. Based on this decomposition, the NTL in the SFAC framework is formally expressed as:

$$NTL = A * (NTL_{fix} + NTL_{variation}) + NTL_{mismatch} + \varepsilon \qquad (1)$$

Where $A * (NTL_{fix} + NTL_{variation})$ is the light from fixed area, $NTL_{fix}$ means the regular light emission from the fixed area, $NTL_{variation}$ represents the changes in NTL that are extremely different from regular patterns due to specific events. $NTL_{mismatch}$ is the light that exists at the areas within the daily changing footprint but outside the fixed area (Fig. 1a); A is the angular effect coefficient that helps correct NTL at different observation angles to the nadir observation. ε denotes residual random noise, which could originate from initial sources or be introduced during the product generation process."

*Reference:*

Hu, Y., Zhou, X., Yamazaki, D., and Chen, J.: A self-adjusting method to generate daily consistent nighttime light data for the detection of short-term rapid human activities, Remote Sens. Environ., 304, 114077, https://doi.org/10.1016/j.rse.2024.114077, 2024.

**Q3:** Adding a scale bar in Fig. 4 and 5 can help us tell if the current cases are solid or not. For example, in Fig. 5, if a 500x500m pixel is much larger than a flyover, the flyover's surrounding light emissions could affect the validation.

**Response:** *Thanks for your valuable comments on revision of these two figures. We have added scale bars in Fig. 4.*

[Figure]

Figure 4. Spatial mismatch correction effectiveness in less-angular effect area within airport cases. (a) Beijing International Airport, (b) Los Angeles International Airport, (c) Sydney International Airport. Google Earth images © 2025 Google LLC.

*Regarding the case of the flyover in Figure 5 of the original manuscript, we agree that the selected flyover is insufficient to demonstrate the effectiveness of spatial mismatch correction accurately. The goal of spatial mismatch correction is to mitigate the influence of neighboring pixels on the target pixel's nighttime light radiance caused by misalignment between the sensor footprint and the final dataset grid. This correction contributes to both a reduction in temporal fluctuations and improved preservation of radiance intensity differences among various lighting objects.*

*To better illustrate the effect of spatial mismatch correction, we selected straight highway segments as case studies. The chosen sections—Tianfu Avenue South Section 2 in Chengdu and the Beijing–Hong Kong–Macao Expressway in Zhengzhou—have no surrounding buildings or other construction land obstructing their view, meaning their lighting mainly comes from streetlights and vehicle headlights, and less angular effect.*

*Given that the spatial resolution of the NTL data is around 500 meters, the pixel width is sufficient to cover the highway width, yet adjacent pixels remain highly susceptible to spatial mismatch effects. A transect analysis across each highway was conducted to evaluate the spatial mismatch correction response (Figure R2). In both the Chengdu and Zhengzhou cases, correction resulted in a more pronounced contrast in radiance intensity between the highway pixels and the non-urban land on both sides, manifested as a narrowing of the peak in the NTL intensity profile.*

*Moreover, due to the uniform distribution of streetlights along these highways, the corrected NTL values of highway pixels are expected to exhibit greater spatial consistency. We therefore quantified the spatial variability (SV) of all highway pixels using the normalized annual mean and standard deviation, calculated as:*

$$SV = \frac{std(N)}{mean(N)}$$

*The SV helps quantify inter-pixel contrast, with lower values indicating lower radiance differences between pixels. After extracting the highway pixels and computing their SV, we observed a decrease in SV following the correction. This result confirms the effectiveness of the spatial mismatch correction. Figure R2 was added to the revised manuscript (Figure 5).*

*We have revised this flyover-related descriptions in 3.3.1 in the new manuscript:*

"In addition, to eliminate the superposition effect of angular effect correction, we selected airports and highways without surrounding obstructions as sample sites to conduct this evaluation for the following reasons: (1) As functional areas with transportation attributes, airports and highways generally do not experience large fluctuations in daily light changes, are less affected by extreme events, and tend to be more

stable in time series. (2) Airports are generally built in open areas, and the highways we selected generally have no surrounding buildings or other construction land obstructing their view. So both of them are less affected by angular effects. To validate the effectiveness of spatial mismatch correction, we calculated the normalized standard deviation both spatially and temporally within the case study area. Furthermore, in the highway case, due to the uniform distribution of streetlights along these highways, the corrected NTL values of highway pixels are expected to exhibit greater spatial consistency. We therefore quantified the spatial variability (SV) of all highway pixels using the normalized annual mean and normalized standard deviation, calculated as:

$$SV = \frac{std(N)}{mean(N)} \qquad\qquad (11)$$

The SV helps quantify inter-pixel contrast, with lower values indicating lower radiance differences between pixels."

*In 4.1:*

"A highway without surrounding obstructions that affect satellite observations can serve as another good sample to validate the effectiveness of spatial mismatch correction. The chosen sections—Tianfu Avenue South Section 2 in Chengdu and the Beijing–Hong Kong–Macao Expressway in Zhengzhou—have no surrounding buildings or other construction land obstructing their view, meaning their lighting mainly comes from streetlights and vehicle headlights, and less angular effect (Chang et al., 2019) (Fig.5). Given that the spatial resolution of the NTL data is 500 meters, the pixel width is sufficient to cover the highway width, yet adjacent pixels remain highly susceptible to spatial mismatch effects. A transect analysis across each highway was conducted to evaluate the spatial mismatch correction response (Fig. 5). In both the Chengdu and Zhengzhou cases, correction resulted in a more pronounced contrast in radiance intensity between the highway pixels and the non-urban land on both sides, manifested as a narrowing of the peak in the NTL intensity profile. Moreover, due to the uniform distribution of streetlights along these highways, the corrected NTL values of highway pixels are expected to exhibit greater spatial consistency. We therefore quantified the SV of all highway pixels. We observed a decrease in SV following the correction in Fig.5. This result confirms the effectiveness of the spatial mismatch correction."

*We replaced Figure 5 with the following:*

[Figure]

Figure R2 (Figure 5). Spatial mismatch correction effectiveness in less-angular effect areas within highway cases. (a) Chengdu Tianfu Avenue, (b) Zhengzhou section of Beijing-Hong Kong-Macao Expressway. Land cover data from the European Space Agency (ESA) WorldCover 10 m 2020 product.

**Q4:** In Fig. 10, what do (a) and (b) refer to?

**Response:** *Thanks for your valuable suggestions. The Fig.10 (a) refers to Comparison of CPSI calculated based on HDNTL with official report data. The Fig.10 (b) refers to Comparison of CPSI calculated based on VNP46A2 with official report data. We have added subtitles to both (a) and (b) in Figure 10.*

[Figure]

Figure 10: Comparison of CPSI calculated based on VNP46A2 and HDNTL with official report data.

**Q5:** I found that all pixels in the 2021-6-9 NTL data of Fuzhou, Fujian, China are 0, as I known Fuzhou is rainy that day. So I am wondering if the method proposed in this study can correct the daily NTL data on a cloudy or rainy day, especially during a continues cloudy/rainy days?

**Response:** *Thank you for your valuable comment. We would like to clarify that our dataset does not fill missing pixels in cloudy/rainy days. The rationale is that nighttime light remote sensing generally reflects human activities, which cannot be accurately predicted or reconstructed during periods of extensive cloud cover. Therefore, there is currently no reliable method to fill in large areas of missing data caused by continuous cloudy or rainy days. These data gaps will not be predicted by our framework to ensure the overall reliability of the nighttime light data we currently produced. Our method is designed to fill only small, spatially or temporally discontinuous gaps in otherwise high-quality data, where interpolation is more likely to be reasonable. To avoid any confusion, we further clarified this point in Section 3.2:*

"Importantly, we do not interpolate missing pixels caused by large and long-persistent clouds as such interpolation is considered unreliable (Román et al., 2018). Our method is specifically designed to fill only small, spatially discontinuous gaps in otherwise high-quality data, where interpolation is more likely to be reasonable. To distinguish small missing holes from those caused by extensive coverage of the cloud mask, we define the missing value hole as a missing value pixel with valid neighboring observations in the spatial windows."

*Additionally, to prevent confusion among end users, we removed images heavily obscured by clouds (percentage of invalid pixels >50%, pixels with constant invalid values, such as water bodies, are not considered in the calculation) from the daily NTL time series. The total daily data for each city in each year included in the HDNTL data is recorded in a .csv file for users to query. Additionally, we provide a flag layer indicating the source of the angular effect reference value and whether interpolation was applied. This flag is encoded as a two-digit integer. The tens digit (values 0, 1, 2, or 3) denotes the source of the nadir-group reference value: 0 for an invalid value, 1 for the mean value of nadir observations from the current year, 2 for the mean value from the current year and its two adjacent years (except for boundary years such as 2012 or 2024, where only one adjacent year is available), and 3 for the annual mean value of all valid observations from the current year. The units digit (values 0 or 1) indicates the interpolation status: 0 for no interpolation and 1 for an interpolated value. The storage structure of this flag is consistent with that of the HDNTL data product.*

**Q6:** What is the maximum/optimal size of holes that the spatiotemporal interpolation can deal with?

**Response:** *Thanks for your question. As stated in Section 3.2* "To distinguish missing value holes from those caused by extensive coverage of the cloud mask, we define the missing value hole as a missing value pixel with valid neighboring observations in the spatial windows." *Therefore, rather than determining the maximum or optimal size of holes based on the interpolation method, we first define a hole as a missing pixel that has at least 4 valid neighboring values within a 3×3 spatial window. Given the relatively coarse resolution of nighttime light data and the significant surface heterogeneity in urban environments, we chose the smallest spatial window (3×3) for spatial interpolation to ensure reliability. For temporal interpolation, we tested single-sided temporal windows ranging from 1 to 8 days, and found that a window size of 5 days yielded the best results in terms of interpolation accuracy and effectiveness.*

*In summary, our spatiotemporal interpolation can handle holes with a maximum size limited to those missing pixels that have at least 4 valid observations within a 3×3 spatial window. Larger or more continuous missing areas, such as those caused by extensive or persistent cloud cover, are not interpolated by our method.*

**Reviewer #2:** This paper presents a valuable contribution to the field of nighttime light remote sensing by addressing the limitations of the VNP46A2 dataset and producing a high-quality daily nighttime light (HDNTL) dataset for 653 cities. The proposed framework for error correction, interpolation, and validation demonstrates significant improvements in data accuracy and reliability. The results are well-supported by comparisons with other datasets and case studies.

**Response:** *Thanks for your constructive suggestions. We have carefully read your comments and thoroughly responded to your concerns. A detailed point-by-point response is listed below.*

**Q1:** I suggest the authors should provide a more detailed explanation for choosing 3 times the standard deviation as the threshold for identifying NTL variation ($NTLvariation$). This threshold lacks a clear justification based on the data characteristics or statistical principles.

**Response:** *Thanks for your valuable suggestion. The threshold of mean+3\*standard deviation($\sigma$) for detecting $NTL_{variation}$ was also adopted from SFAC method by Hu et al (2024). This criterion is statistically rigorous, as it aligns with Pukelsheim's 3$\sigma$ rule (1994), which states that for any unimodal distribution, $\geqslant$99% of data falls within 3$\sigma$ of the mean. From empirical validation, we randomly sampled daily NTL sequences (2022-2023) from two pixels in Beijing and plotted their histograms with fitted normal curves (Fig.R3). The results confirm that the daily NTL series **follows a normal distribution.** Approximately **99% of observations lie within ±3$\sigma$**, while values beyond this threshold (outliers) could be considered as abrupt NTL variations (e.g., lighting disruptions or festive events).*

*Thus, this threshold balances statistical robustness and practical applicability. We appreciate your feedback and have added these clarifications to the manuscript in Section 3.1.*

"Therefore, the A-average filter processes firstly identified and removed the $NTL_{variation}$ using a 3 times standard deviation threshold relative to the annual mean light radiance (Hu et al., 2024; Pukelsheim, 1994) ."

[Figure]

Figure R3: Normal distribution fitting of the daily NTL series in a year.

*Reference:*

Hu, Y., Zhou, X., Yamazaki, D., and Chen, J.: A self-adjusting method to generate daily consistent nighttime light data for the detection of short-term rapid human activities, Remote Sens. Environ., 304, 114077, https://doi.org/10.1016/j.rse.2024.114077, 2024.

Pukelsheim, F., 1994. The three sigma rule. Am. Stat. 48, 88–91.

**Q2:** The use of mean NTL radiance for angular effect correction may introduces some uncertainty. Maybe the authors can consider exploring methods such as fitting a model to the relationship between NTL pixels for each VZA. This could provide a more accurate and robust correction, particularly for complex urban environments.

**Response:** *Thank you for your constructive suggestion. We agree that the angular effect can be modeled by linear or non-linear models (Tan et al., 2022; Li et al., 2019). In our framework, we used the near-nadir observations as the reference and correct observations from other angles to the near-nadir reference, mainly because two reasons: (1) for each pixel, the satellite*

*observation angles throughout a year have **16 discrete VZA groups**. As shown in Figure R3, the differences between groups are much larger than the variations within each group. Therefore, the annual variation in observation angles for a given pixel is not particularly complex, and the group-based correction approach is sufficient to address the angular effect. (2) pixel-based model fitting is time consuming for data generation in large areas. Angular correction based on angle grouping provides a pixel-adaptive solution to angular effects and is relatively efficient.*

*We clarified this point in the revised manuscript in 3.1:*

"The anisotropic characteristics of NTL and their relationship with satellite viewing angles have been well documented and modeled (Tan et al., 2022; Li et al., 2019). The S-NPP satellite adopts a sun-synchronous polar orbit and completes a revisit cycle every 16 days. This orbital configuration ensures that each ground location maintains essentially the same VZA observation sequence within the 16-day cycle in a given year. For each pixel, the satellite observation angles throughout a year fall into 16 discrete VZA groups. The differences between groups are much larger than the variations within each group. Consequently, the annual variation in observation angles for a given pixel is not particularly complex. Moreover, pixel-based model fitting is time consuming for data generation in large areas. Therefore, we still follow the SFAC method of Hu et al. (2024), as the group-based correction approach is sufficient to address the angular effect."

*References:*

Tan, X., Zhu, X., Chen, J., and Chen, R.: Modeling the direction and magnitude of angular effects in nighttime light remote sensing, Remote Sens. Environ., 269, 112834, https://doi.org/10.1016/j.rse.2021.112834, 2022.

Li, X., Ma, R., Zhang, Q., Li, D., Liu, S., He, T., & Zhao, L.: Anisotropic characteristic of artificial light at night – Systematic investigation with VIIRS DNB multi-temporal observations, Remote Sens. Environ., 233, 111357, https://doi.org/10.1016/j.rse.2019.111357, 2019.

**Q3:** The paper only uses one flyover site to validate the effectiveness of spatial mismatch correction. It is recommended to include additional examples to strengthen the evidence of the correction's generalizability.

**Response:** *Thanks for your constructive advice. Related concerns have been thoroughly addressed in our responses to the other reviewers, and we believe these responses also apply to your questions. Please refer to our response to Reviewer 1's Comment #3 for details.*

**Q4:** The sudden increase in RMSE for the temporal window size of 4 in Figure 7 requires further investigation and explanation.

**Response:** *Thanks for your constructive advice. The sudden increase in RMSE observed for a temporal window size of 4 in Figure 7 was indeed caused by the previous spatial interpolation step, which interpolates any pixels with the number of valid pixel > 0 in a 3\*3 window. This setting resulted in the inclusion of pixels along coastlines, rivers—areas where both spatial and temporal reference values are often very limited. The interpolation of these edge pixels introduced significant errors, thereby inflating the overall RMSE and MAE.*

*To mitigate this interpolation error, we have revised the spatial interpolation criterion to the number of valid pixel > 4 in the spatial window, meaning that a missing-value pixel must now have more than half of the 3×3 window containing valid values to be considered for interpolation. This adjustment effectively excludes unnecessary pixels near water bodies and region boundaries, substantially reducing noise and improving interpolation accuracy.*

*Based on the modified spatial interpolation criteria, we have recalculated the RMSE, MAE and relative temporal weights for temporal window sizes 1 through 8. We also revised the scatter plot for temporal window sizes 5. The corresponding descriptions in Section 3.2 and 4.3 have also been updated accordingly.*

[Figure]

Figure 1: Interpolation accuracy test. (a) The RMSE and MAE of interpolation by different time windows, (b) The relative temporal weight of different time windows, (c) The comparison between reference data and its interpolation by the 5-day temporal window.

*We added the related description in the new manuscript in 3.2:*

"To avoid overfilling edge pixels near coasts, lakes, or ROI boundaries, we only filled missing-value pixels that had at least four valid reference pixels within the 3×3 window."

*We revised the related results in the new manuscript in 4.3:*

"According to Fig. 7a, when the single-sided window was set to 5, the RMSE and MAE of the interpolation were lowest……As shown in Fig. 7c, the linear regression $R^2$ between the reference value and the interpolation value was 0.98, the Pearson correlation coefficient (r) was 0.99 (p< 0.001), the RMSE was 2.64, and the MAE was 1.24, indicating the high accuracy and reliability of the comprehensive interpolation method. After the hole interpolation implementation, all cities' valid pixels increased by about 2%."

**Q5:** The paper mentions the "mutation signal recognition threshold" without providing a clear definition.

**Response:** *Thanks for your constructive advice. We have clarified the description in Section 3.3.3 as follows:*

"When the absolute value of DA is greater than 3 (usually used as the threshold for mutation signal detection), it suggests that the value of this event deviates statistically significantly from the normal distribution and may be an outlier or extreme value (Pukelsheim, 1994)."

*Reference:*

Pukelsheim, F., 1994. The three sigma rule. Am. Stat. 48, 88–91.

**Reviewer #3:** This paper utilizes an improved SFAC method to correct the VNP46A2 NTL product, specifically addressing the VZA effect and mismatches. These corrections enhance the stability of the NTL time series while preserving critical anomalies caused by events such as natural disasters or holidays. The overall structure of the manuscript is clear; however, it lacks explanation of some key procedures and a deeper discussion of results. Therefore, my recommendation is a major revision. My specific comments are as follows:

**Response:** *Thanks for your constructive suggestions. We have carefully read your comments and thoroughly responded to your concerns. A detailed point-by-point response is listed below.*

**Q1:** On page 5, line 125, the phrase "Grow a square with... is less than 1" is unclear. What exactly does "less than 1" refer to in this context

**Response:** *Thanks for your insightful suggestion. In our process, we start from the central coordinates of each city and iteratively expand a square region. The expansion continues until **the ratio of** the total urban core area to the combined suburban and rural area within the square **less than 1**. In other words, "less than 1" means that the total urban core area is smaller than the combined suburban and rural area. To avoid any confusion, we have revised the description of this process in Section 2.1 as follows:*

"Starting from the central coordinates of each city, a square region is iteratively expanded until the total urban core area is smaller than the combined suburban and rural area within the square. The expansion stops once this condition is met, and the final square boundaries are recorded as the city's ROI."

**Q2:** On page 5, line 127, the authors mention that in highly urbanized areas, overlapping convex hulls may emerge. To address this, a manual adjustment method is employed. However, the manuscript does not specify the criteria for this manual adjustment, nor does it describe the resulting delineation of urban areas after resolution.

**Response:** *Thanks for your constructive advice. While our initial approach may suggest the possibility of overlapping regions in highly urbanized areas, we did not perform manual adjustments specifically to resolve overlaps between ROIs. Instead, the key manual*

*adjustment addressed the following issue: for cities where the initial ROI did not fully encompass the contiguous urban core area, we manually adjusted the ROI to guarantee complete coverage; for special cases involving urban agglomerations (e.g., Los Angeles-Long Beach-Santa Ana), we also performed visual inspections and manual adjustments to ensure that the ROI covered the core built-up areas of all cities within this urban agglomeration. To clarify this process, we have revised the relevant descriptions in 2.1 as follows:*

"The expansion stops once this condition is met, and the final square boundaries are recorded as the city's ROI. For cities where the initial ROI did not fully encompass the contiguous urban core area, we manually adjusted the ROI to guarantee complete coverage. For special cases involving urban agglomerations (e.g., Los Angeles-Long Beach-Santa Ana), we also performed visual inspections and manual adjustments to ensure that the ROI covered the core built-up areas of all cities within this urban agglomeration. In regions where cities are in close proximity, some ROIs may overlap, as illustrated by the close-up of cities in the Yangtze River Delta in China (see Fig. 2); no further adjustments were made in these cases."

**Q3:** On page 6, line 149, the data used to validate the correction results is from Tianjin City, covering the period from January 25 to February 21, 2022. Clearly, relying on a single region and a one-month span is not robust. Is there a way to increase the volume of validation data? If acquiring more data from SDGSAT is not feasible, could alternative validation methods be considered?

**Response:** *Thanks for your constructive advice. First, due to observational quality issues, many SDGSAT-1 data phases are not publicly available. Of the publicly available data, most are severely cloud-contaminated and noisy. As a result, the clear, contemporaneous two-phase imagery from Tianjin was among the only suitable and accessible reference data available at the time of this study. We agree that expanding the validation scope would strengthen the results, and we are actively seeking additional open SDGSAT-1 datasets or comparable high-resolution nighttime light data for future work.*

*Secondly, we understand the reviewer's concern regarding the one-month span. We would like to clarify that the objective of this validation was not to track a specific, gradual change process over time, but rather to use a "changed scenario" to test HDNTL's "change detection capability". Nighttime light exhibits day-to-day variability. Consequently, a monthly interval is sufficient to capture statistically significant changes, thereby providing a valid scenario for*

*assessing detection capability. Our core objective is to demonstrate that the high consistency in change patterns between HDNTL and SDGSAT-1 over this period confirms the effectiveness of HDNTL in capturing relative changes in nighttime light. The capability of change detection is an inherent property of the data product itself and is independent of the specific temporal frequency (daily, weekly, or monthly) at which it is ultimately applied.*

**Q4:** On page 8, line 182, the authors claim that "it is reasonable to assume that the annual lowest light intensity is close to the NTLfix." However, the rationale behind this assumption is not provided. Why wouldn't the annual mean light intensity be a better approximation to NTLfix?

**Response:** *Thanks for your insightful question. Our approach follows the methodology proposed by Hu et al. (2024) in the SFAC method. The key idea is that the fixed area, which represents regular light emissions (NTL_fix), is inherently smaller than the daily varying sensor footprint (i.e., the fixed area of a pixel is smaller than daily footprint). As a result, the observed nighttime light (NTL) signal at any given time can be decomposed as follows (Hu et al., 2024):*

$$NTL = A * (NTL_{fix} + NTL_{variation}) + NTL_{mismatch} + \varepsilon$$

*If we were to use the annual mean light intensity as an estimate for $NTL_{fix}$, the value would inevitably include an additional portion of $NTL_{mismatch}$. This would lead to an overestimation of the stable light emissions in fixed area. By contrast, using the annual minimum light intensity helps to minimize the influence of $NTL_{mismatch}$. Except in cases of abrupt light reduction events (such as power outages), the lowest observed value in a year is most likely to reflect periods when only the stable, persistent sources are contributing to the signal, with minimal interference from transient or mismatched sources. While it is possible that some residual mismatch error remains, in the absence of ground-truth data, the annual minimum provides the closest achievable approximation to $NTL_{fix}$. We have revised the relevant section of the manuscript to clarify this reasoning as follows:*

"This approach operates under the principle that the fixed area (representing regular light emissions) is inherently smaller than the daily varying sensor footprint. Accordingly, the annual lowest light intensity within a given year is used to approximate the stable light intensity coming from the fixed area ($NTL_{fix}$), except in cases of abrupt light reduction events such as power outages ($NTL_{variation}$)."

*References:*

Hu, Y., Zhou, X., Yamazaki, D., and Chen, J.: A self-adjusting method to generate daily consistent nighttime light data for the detection of short-term rapid human activities, Remote Sens. Environ., 304, 114077, https://doi.org/10.1016/j.rse.2024.114077, 2024.

**Q5:** On page 9, line 204, the authors divide the VZA into 16 groups. How exactly was this division made? Was the 0–60° range evenly partitioned into 16 intervals? Although the grouping is said to be based on the 16-day revisit cycle, in practice, very similar VZA values can occur within the same cycle. For example, in a single cycle, if one day has a VZA of 54° and another day 58°, would these be grouped together?

**Response:** *Thanks for your constructive comment. We did not divide the 0–60° VZA range into 16 equal intervals. Instead, the grouping was strictly based on the 16-day revisit cycle of the satellite. For example, data from the 1ˢᵗ, 17ᵗʰ, 33ʳᵈ …. days were grouped together, because these observations from the same orbit and have very similar VZA.*

*To illustrate this, Figure R4 shows the VZA variation for a randomly selected pixel over one year, as well as the differences within and between groups. For example, in Figure R4(a), the circled VZA values of 54.93° and 53.12° occur within the same cycle and are very close in value. However, as shown in Figure R4(b), even though these two VZA values are similar, their differences with other values within their respective groups are even smaller. Figure R4(c) further demonstrates that the maximum intra-group difference is still smaller than the minimum inter-group difference. Therefore, grouping based on the day sequence within the 16-day cycle is reasonable.*

*We have revised the relevant descriptions of the manuscript to clarify this grouping method and its rationale in 3.1:*

"Based on this characteristic, we divided the daily NTL time series for each year into 16 groups according to the day sequence within the 16-day cycle. For each group, we calculated the mean NTL radiance. The group with near-nadir observations (VZA < 6º) was designated as the reference group for angular effect correction. The angular effect coefficients and correction value of the remaining groups were calculated as follows:"

[Figure]

Figure R4: VZA groups and its intra-group and intergroup differences. (a) Daily VZA changes in one year, (b) Sorted distribution of VZA in one year, (c) Within-group and between-group differences.

**Q6:** On page 9, line 227, the authors apply an inverse distance weighting to non-central pixels in a 3×3 window. However, in such a window, the distances to the center pixel are identical for all non-central pixels, resulting in identical weights. What then is the significance of applying this weight?

**Response:** *Thank you for your thoughtful question. We retain the term $\sum_n^N \omega_n$ in Eq.(4) because it will later be used to calculate a valid weight ratio, which in turn facilitates the calculation of the relative spatial weight. The valid weight ratio represents the weighted proportion of reference information available within a given spatial or temporal window. Therefore, even though the assignable weight for each surrounding pixel is the same spatially, we retain in Eq.(4) to better illustrate the calculation process of Eq.(6).*
*We have revised this in 3.2:*
"The interpolation process is as follows:

$$NTL_{is} = \sum_n^N \omega_n NTL_n \tag{4}$$

$$NTL_{it} = \sum_m^M \omega_m NTL_m \tag{5}$$

$$W_{is} = \frac{\sum_n^N \omega_n \cdot valid_n}{\sum_n^N \omega_n \cdot 1} \tag{6}$$

$$W_{it} = \frac{\sum_m^M \omega_m \cdot valid_m}{\sum_m^M \omega_m \cdot 1} \tag{7}$$

$$valid_n = \begin{cases} 1, & NTL_n \ is \ valid \\ 0, & NTL_n \ is \ invalid \end{cases} \tag{8}$$

$$NTL_i = a_{is} NTL_{is} + a_{it} NTL_{it} \tag{9}$$

$$a_{is} = \frac{W_{is}}{W_{is}+W_{it}}, a_{it} = \frac{W_{it}}{W_{is}+W_{it}} \tag{10}$$

Where $NTL_{is}$ is the spatial fill value for pixel $i$, $n$ is the $n$-th pixel in a 3×3 window centered on pixel $i$, ……; $W_{is}$ and $W_{it}$ represent the valid weight ratios in the spatial and temporal interpolation processes, respectively. ……$NTL_{variation}$ is also excluded during interpolation and added back after interpolation."

**Q7:** On page 11, line 270, the authors select the Liuyang Firework Festival as a case study to test whether their method suppresses genuine NTL anomalies. However, aerosols from fireworks can significantly affect NTL intensity, especially during windless nights. Have the authors considered this potential confounding factor?

**Response:** *Thank you for your valuable comment. We agree that aerosols can affect NTL intensity, especially under windless conditions. However, in our case study, the observed increase in nighttime light is not solely attributable to the fireworks themselves. As mentioned in the manuscript, the Firework Festival attracts a large number of vendors and visitors, and the festivities continue late into the night, resulting in a significant increase in overall nighttime activity and lighting. This leads to a much higher NTL signal on the festival day compared to regular days.*

*While aerosols may partially attenuate the NTL signal, the overall increase in brightness during the festival is still clearly detectable and distinguishable from normal variations. Therefore, we believe that the potential confounding effect of aerosols does not prevent the identification of genuine NTL anomalies associated with the festival.*

**Q8:** In Figure 5, the selected flyover region seems unsuitable for validation purposes, as it is surrounded by many tall buildings within a 500-meter radius, which could introduce substantial VZA effects.

**Response:** *Thanks for your constructive advice. Related concerns have been thoroughly addressed in our responses to the other reviewers, and we believe these responses also apply to your questions. Please refer to our response to Reviewer 1's Comment #3 for details.*

**Q9:** In Figure 5, the IR values derived from the original VNP46A2 data and from the HDNTL product are almost identical. Does this imply that the proposed method is ineffective in this case? This may support the previous concern regarding the selection of this flyover site.

**Response:** *Thanks for your constructive advice. Related concerns have been thoroughly addressed in our responses to the other reviewers, and we believe these responses also apply to your questions. Please refer to our response to Reviewer 1's Comment #3 for details.*

**Q10:** In the Beirut port explosion case shown in Figure 9, why does the HDNTL product not show the NTL intensity on the day of the explosion as the lowest of the year? The authors should provide an explanation for this discrepancy.

**Response:** *Thank you for your insightful comment. First, it is not necessarily expected that the NTL intensity on the day of the Beirut port explosion would be the lowest value of the year. After the explosion, emergency response and rescue operations were carried out in the affected area, and the site did not undergo effective reconstruction during a long time even until now. As a result, there may have been subsequent days with even lower NTL values due to prolonged power outages, restricted access, or further infrastructure degradation.*

*Second, the primary goal of our correction is to enhance the detectability of abrupt events such as the explosion. This improvement in event detection is reflected in the DA values, which demonstrate that our HDNTL product more clearly highlights the occurrence and impact of the explosion compared to the original data.*

**Reviewer #4:** This study generated daily nighttime light (NTL) data for 600 more global large cities. Since NTL is the only satellite remote sensing data to show the nighttime activities on Earth, this dataset is highly desirable for urban studies. Spatial mismatch and angular effect are two major problems which lead to big uncertainties in the daily NTL time series, so this study improves the data quality of daily NTL by correcting these two effects. In addition, it fixes the small holes in cloud-free images to further improve the data availability. Generally speaking, the methodology used by this study is correct and the produced data has high value in applications. I have some specific comments for the authors to further improve this study.

**Response:** *Thanks for your constructive suggestions. We have carefully read your comments and thoroughly responded to your concerns. A detailed point-by-point response is listed below.*

**Q1:** Abstract: I would suggest using r instead of $r^2$ to report the similarity between interpolated and reference data.

**Response:** *Thank you for your valuable suggestion. We agree that using Pearson's r is indeed more appropriate for directly assessing the linear relationship and directionality between interpolated and reference data, as it avoids potential misinterpretations associated with $R^2$ in this context. We have revised this part in Abstract and Section 4.3 accordingly and appreciate your observation:*
*In Abstract:*
"The spatiotemporal interpolation of missing data holes shows high similarity with the reference data, as indicated by a Pearson correlation coefficient (r) of 0.99, and it increased the valid pixels of all cities by about 2%. "
*In 4.3:*
"As shown in Fig. 7c, the linear regression $R^2$ between the reference value and the interpolation value was 0.98, the Pearson correlation coefficient (r) was 0.99 (p< 0.001), the RMSE was 2.64, and the MAE was 1.24, indicating the high accuracy and reliability of the comprehensive interpolation method."

**Q2:** Introduction: The authors mentioned that the daily NTL is affected by external factors such as measurement errors and retrieval uncertainties. It is not clear what measurement errors are. In addition, NTL is easily affected by clouds, which is a big problem in applications. I understand that cloud removal is not done by this study, but the authors should clarify this at the very beginning to avoid misleading readers.

**Response:** *Thank you for raising these important points. We acknowledge that the terms "measurement errors" and "retrieval uncertainties" require further clarification to avoid ambiguity. Measurement errors primarily refer to geometric and observational limitations, including spatial mismatches (caused by discrepancies between sensor footprints and data grid resolution) and angular effects (due to variations in satellite viewing angles). Retrieval uncertainties stem from algorithmic challenges in the VNP46A2 product resulting in missing data, as highlighted in NASA's Black Marble Nighttime Lights Product Suite Algorithm Theoretical Basis Document (ATBD). For example, "Flagging of suspect VIIRS Cloud Mask (VCM) detections is done in the Lunar BRDF correction process (nighttime branch), which outputs a poor-quality mandatory QA flag when the VNP46 algorithm fails to produce a reliable NTL result", and "Results from benchmark #4 illustrate how the performance of the nighttime VCM varies significantly depending on factors such as moon-illumination conditions, surface albedo, as well as atmospheric, climatic, and geographic conditions". Therefore, we conclude the spatial mismatch, angular effect and missing data holes to "measurement errors" and "retrieval uncertainties".*

*We acknowledge that the use of overly summarized terminology in the Introduction section, without sufficient explanation of the specific errors, may lead to misunderstandings among readers. Therefore, we have comprehensively revised the Introduction to provide a clearer and more logical articulation of these issues.*

"With advances in satellite remote sensing, higher temporal and spatial resolution NTL data have become available, enabling more detailed and timely monitoring of human activities......Among the current NASA Black Marble product suites, the VNP46A2 dataset has been daily moonlight- and atmosphere-corrected based on daily at-sensor TOA nighttime radiances (VNP46A1). Despite these improvements, several challenges remain for daily NTL products, especially regarding data quality and reliability (Wang et al., 2021; NASA, 2018). Notably, spatial observational coverage mismatch (hereafter spatial mismatch, i.e., misalignment between satellite footprint and output grid) (Campagnolo et al., 2016; Román et al., 2018; Wolfe et al., 1998), angular effect (i.e., systematic variations in observed NTL radiance with changing viewing zenith angle) (Tan et al., 2022) and missing data holes (i.e., scattered pixels flagged as low-quality or missing values in cloud-free images) have all been show to introduce

significant uncertainties into daily NTL series. These issues can lead to spurious fluctuations or gaps in the observed NTL intensity that do not reflect actual changes in human activity, thereby limiting the accuracy and reliability of short-term human activity monitoring based on daily NTL data (Fig.1; Hu et al., 2024; Wang et al., 2021).

Specifically, the spatial mismatch error arises when the 740m DNB view footprint misaligns with the 500m gridded output pixels, leading to brightness measurement errors (Hu et al., 2024) (Fig. 1a), reported as the relatively large temporal daily radiance variation observed (Román et al., 2018) (Fig. 1b)."

**Q3:** The authors argued that the spatial mismatch, angular effect, and missing data holes introduce significant uncertainties in daily NTL. May I know how much of the uncertainties were caused by these three issues? Some examples can be used to explain this. It can help readers understand the necessity of correcting these issues.

**Response:** *Thank you very much for your constructive comments. Quantifying the exact uncertainties is challenging due to these errors' interactions. Furthermore, the necessity of correcting these errors has been extensively discussed and validated in previous studies, which have demonstrated their significant impact on the accuracy and reliability of daily NTL data. Therefore, we did not repeat the detailed quantification in our manuscript.*

*We greatly appreciate your suggestion. In response, we have carefully revised the Introduction to improve its readability and logical flow, and we have provided a clearer illustration of the spatial mismatch error, angular effect, and missing data holes in Figure 1 to help readers better understand the sources and manifestations of these uncertainties.*

*References:*

Campagnolo, M. L., Sun, Q., Liu, Y., Schaaf, C., Wang, Z., & Román, M. O. (2016). Estimating the effective spatial resolution of the operational BRDF, albedo, and nadir reflectance products from MODIS and VIIRS. Remote Sensing of Environment, 175, 52–64. https://doi.org/10.1016/j.rse.2015.12.033

Wolfe, R. E., Roy, D. P., & Vermote, E. (1998). MODIS land data storage, gridding, and compositing methodology: Level 2 grid. IEEE Transactions on Geoscience and Remote Sensing, 36(4), 1324–1338. https://doi.org/10.1109/36.701082

**Q4:** Study area: The authors talked about the situation of 2030, but they used the 2025 population to select cities as their study areas. It is a little bit confusing.

**Response:** *Thank you for your insightful comment. To avoid any confusion, we have removed the sentence referring to the 2030 projection. This change does not affect the overall meaning or integrity of the paragraph.*

**Q5:** Page 5 line 125: Most cities' built-up areas are within the ROI. How many cities are not completely covered by the ROI? If a city has a large area not covered by the ROI, was it manually adjusted?

**Response:** *Thank you for your constructive comment. During the ROI generation process, we carefully examined each city's ROI by comparing the city boundary and the extent of the contiguous urban core area. We ensured that the ROI for every city adequately covers the contiguous urban core area radiating from the city center. For cities where the initial ROI did not fully encompass the contiguous urban core area, we manually adjusted the ROI to guarantee complete coverage. As a result, all 653 cities in our study have ROIs that fully cover their respective built-up areas. To avoid any confusion, we have revised the description in Section 2.1 as follows:*

"The extent of each city, defined as its Region of Interest (ROI), is adaptively determined based on the GHS Degree of Urbanisation Classification (GHS-DUC) dataset (Schiavina et al., 2021). Our goal is to ensure that the ROI for every city adequately covers the contiguous urban core area radiating from the city center. The process for delineating each ROI is as follows: (1) The GHS-DUC data are reclassified into three categories: urban core area ("city", "dense town", and "semi-dense town" in GHS-DUC ), suburban and rural area ("suburbs or peri-urban area", "village", "dispersed rural area", and "mostly uninhabited area" in GHS-DUC ), and non-land area (including oceans and inland water bodies, categorized as "not on land" in GHSL); (2) Starting from the central coordinates of each city, a square region is iteratively expanded until the total urban core area is smaller than the combined suburban and rural area within the square. The expansion stops once this condition is met, and the final square boundaries are recorded as the city's ROI.  For cities where the initial ROI did not fully encompass the contiguous urban core area, we manually adjusted the ROI to guarantee complete coverage. For special cases involving urban agglomerations (e.g., Los Angeles-Long Beach-Santa Ana), we also performed visual inspections and manual adjustments to ensure that the ROI covered the core built-up areas of all cities within this urban agglomeration. In regions where cities are in close proximity, some ROIs may overlap, as illustrated by the close-up of cities in the Yangtze River Delta in China (see Fig. 2); no further adjustments were made in these cases."

**Q6:** Page 9 line 205: Please clarify how to handle the situation if clouds contaminated the near-nadir observations.

**Response:** *Thanks for your insightful question. The situation where the near-nadir observations cannot be collected exists. Using 2024 for example, we calculated the mean value "Nadir_observation_mean" of the nadir observation group for NTL images of 653 cities in 2024 after spatial mismatch correction. We then computed the proportion of invalid pixels in each city's "Nadir_observation_mean" image relative to the entire image. As shown in the Fig. R5, outliers were identified based on quartiles, revealing that 22 cities had more than 5.6% invalid nadir observations in their 2024 nighttime light images, with 5 cities exceeding 30%.*

[Figure]

Figure R5. The invalid nadir observation ratio of all cities in 2024.

*Taking Chengdu, which had an invalid pixel rate of 62%, as an example, we illustrate our processing method. In the Fig. R6, the "1st reference" refers to the primary reference image, which is the average image of the near-nadir observation group for the target year. It was found that 62% of the pixels in Chengdu's 2024 near-nadir observations did not accumulate more than 3 days of near-nadir observations, so these pixels were deemed invalid references. For pixels with more than 3 days of near-nadir observations, we retained their reference values and generated Valid Reference Mask 1. The "2nd reference" refers to the average image of all near-nadir observation groups from the target year and its two adjacent years (for years such as 2012 or 2024, only the target year and one adjacent year are included). The "3rd reference" refers to the average image of all observations from the target year. In generating the complete nadir observation reference, we prioritized values from the 1st*

*reference, followed by the 2nd and 3rd references. Therefore, we first obtain the "Filled Nadir Reference 1" based on the 1st and 2nd references, and then derive the final "Filled Nadir Reference 2" based on the 3rd reference. The "Filled Nadir Reference 2" will serve as the reference image for the final angular effect correction. The source of nadir reference of each pixel in "Filled Nadir Reference 2" will be recorded in the "Nadir Reference Mask", where:1 indicates that the pixel was corrected based on nadir observations from the 1st reference, 2 indicates that the pixel was corrected based on nadir observations from the 2nd reference, 3 indicates that the pixel was corrected based on nadir observations from the 3rd reference. 0 indicates Nan.*

*We have added a description of this issue to the manuscript in 3.1:*

"To ensure the reliability of the near-nadir reference value and prevent it from being unduly uncertain by an insufficient number of observations, we established a tiered filling strategy for constructing the angular effect reference layer. Specifically, the near-nadir observation group of the target year is used as the primary reference only if it contains more than three days of valid observations. Otherwise, we extend the temporal window to include near-nadir observations from both the target year and its two adjacent years (except for boundary years such as 2012 or 2024, where only one adjacent year is available), and calculate the average value. If the combined observations still do not exceed three days, the annual mean of all observations from the target year—regardless of viewing angle—is used as the fallback reference to guarantee completeness of the reference layer. The data source of each pixel in the final nadir reference image is systematically recorded in the Nadir Reference Flag, with values of 1, 2, and 3 indicating the primary (target year near-nadir), secondary (multi-year near-nadir), and tertiary (target year annual mean) references, respectively."

[Figure]

Figure R6: The process of generating a complete nadir observation reference

**Q7:** Page 9 line 215: These spatiotemporal methods have been developed. Why not directly use them for interpolating holes?

**Response:** *Thanks for your insightful question. We address this question from both operational efficiency and accuracy perspectives:*

1)*Existing spatiotemporal interpolation methods for NTL time series may be overly complex for our task. Methods like Tan et al.'s and Hao et al.'s rely on identifying pixel pair and interpolating gaps based on their temporal profiles. Although this ensures high reliability for single-date interpolation, it could be computationally inefficient for large-scale daily gap-filling. Since our goal is to fill small, isolated gaps (rather than extensive missing regions), we prioritize a simpler yet reliable approach.*

2)*Our method strategically integrates spatial and temporal information. Validation confirms that a 5-day single-window approach achieves the optimal balance—providing sufficient temporal context while minimizing errors. Furthermore, its*

*lightweight implementation on Google Earth Engine (GEE) enables rapid processing, making it operationally feasible for practical applications.*

**Q8:** Figure 5: Only one example of a flyover case was shown. It is better to show more cases. I am not sure whether flyover has no angular effect.

**Response:** *Thanks for your constructive advice. Related concerns have been thoroughly addressed in our responses to the other reviewers, and we believe these responses also apply to your questions. Please refer to our response to Reviewer 1's Comment #3 for details.*

---

## Author Response (AR2)

**Response to Reviewers' Comments**

We would like to express our sincere gratitude to the editors and reviewers for their valuable comments and suggestions, which have greatly improved our manuscript and dataset, making it acceptable for *Earth System Science Data*. In this round, we thank the topic editor for these correction comments. We have carefully reviewed the comments and made the necessary revisions. The changes in the revised manuscript were highlighted in blue.

Q1: Remove the word "holes" in the phrase "missing data holes" in the abstract and elsewhere. This is because "missing data holes" sounds like the "data holes" are missing. Alternatively, you can say "data holes", and remove the word missing. This is a mandatory fix.

**Response:** Thank you for this correction. We have replaced "missing data holes" and "missing holes" with "data holes" throughout the text and Figure 1 to enhance clarity and avoid misunderstanding.

Figure 1: An illustration of spatial mismatch error, angular effect error and data holes of VNP46A2. (a) Error source: Schematic diagram illustrating the existence of spatial mismatch and angular effects, (b) The instability of daily NTL series, (c) Angular effect: The changing trend of daily NTL intensity along with VZA changes within one year, (d) Data holes existence in cloud-free NTL image of Beijing on October 1, 2018, (e) The percentage of data holes in different NTL intensity groups calculated based on annual averages.

**Q2:** HDNTL and all other abbreviations should be defined during the first term of use. There are many abbreviations in this manuscript and they all need to be written out at first use.

**Response:** Thank you for this important reminder. We have carefully reviewed the entire manuscript and ensured that all abbreviations, including HDNTL, are now properly defined at their first occurrence.

**Q3:** Introduction Line 42: "The Day/Night Band (DNB)...is carried". This sentence is confusing. Isn't a band a part of the remotely sensed data? Are you talking about a sensor instead? Please revise.

**Response:** Thanks for your insightful comment. You are correct that the Day/Night Band (DNB) is a key sensor of the Visible Infrared Imaging Radiometer Suite (VIIRS), on board the Suomi-National Polar-orbiting Partnership (S-NPP) and Joint Polar Satellite System (JPSS) satellite platforms. We have corrected this unclear description on Line 42:

"The Day/Night Band (DNB) sensor of the Visible Infrared Imaging Radiometer Suite (VIIRS), is carried aboard the Suomi National Polar-orbiting Partnership (S-NPP) and Joint Polar Satellite System (JPSS) satellites."

**Q4:** "challenging since the discrepancy is random from day to day." -> What does this mean? The word 'random' should be used very carefully—do you mean it's hard to predict? And why?

Response: Thank you for raising this important point. We agree that the term "random" was imprecise, so we have revised the sentence and clarified the reason as follows: "The Black Marble dataset grid 740m DNB observations to 15 arcsec geographic latitude/longitude pixels (approximately 500m), resulting in gridding artifacts (Wang et al., 2021; Hu et al., 2024) (Fig. 1a). In addition, the geolocation accuracy of NASA VIIRS swath products has a mean residual of 2 m with a root mean square error of less than 70 m along track and 60 m along scan direction (Wolfe et al., 2013). Both the geolocation uncertainty and gridding artifacts result in a spatial mismatch, which is reported as the relatively large temporal daily NTL radiance variation (Román et al., 2018) (Fig. 1b). Estimating the error using any model or formula is challenging since the discrepancy varies from day to day."

**Q5:** Remove capitals from here: A Self-adjusting method featuring Filter and Angular effect Correction (SFAC).

**Response:** Thanks for your suggestion. We have revised "A Self-adjusting method featuring Filter and Angular effect Correction (SFAC)" to use lowercase letters.

**Q6:** The term Region of Interest (ROI) is confusing. The reader community of this journal understands that a city sprawls, and that the study area includes the sprawl. ROI usually means Return on Investment. Please replace with 'region', or 'urban region'.

**Response:** Thanks for your valuable suggestion. We agree that the abbreviation ROI is prone to confusion with "Return on Investment," particularly for a multidisciplinary readership. We have replaced all instances of "Region of Interest (ROI)" with the clearer and more appropriate term "urban region" throughout the manuscript to accurately reflect the dynamic nature of our study areas.

**Q7:** Please watch for extra spaces, e.g. "in GHS-DUC")" and elsewhere.

**Response:** Thank you for your meticulous review. We have carefully checked the entire manuscript and corrected all instances of extra spaces.

**Q8:** Reviewer 3's comment "Q1: On page 5, line 125, the phrase "Grow a square with... is less than 1" is unclear. What exactly does "less than 1" refer to in this context" requires better explanation. Accordingly, on line 152: "a square region" is not well-defined. What do you mean by this? Without exact description of size and method, someone would not be able to replicate your study, and this aspect if it is important.

**Response:** Thank you for highlighting this lack of clarity. We have revised the method description as:

"Beginning with the central coordinates of each city, as sourced from the United Nations World Urbanization Prospects (WUP): The 2018 Revision, we defined an initial square region with a side length of 0.1 decimal degrees. This square was progressively expanded using a fixed step size of 0.05 decimal degrees. At each step, the suburban and rural area was compared to that of the urban core within the current square. The square expansion continued until the suburban and rural area exceeded the urban core area, and the final square region was adopted as the urban region of each city."

**Q9:** Line 239: This sentence isn't clear: "For each pixel, the satellite observation angles throughout a year fall into 16 discrete VZA groups."

**Response:** Thanks for your constructive comment. We have revised the description to improve clarity:

"The anisotropic characteristics of NTL and their relationship with satellite viewing angles have been well documented and modeled (Tan et al., 2022; Li et al., 2019). The S-NPP satellite, which operates in a sunsynchronous polar orbit with a 16-day revisit cycle, observes each ground location under a recurring sequence of View Zenith Angles (VZAs). This orbital pattern ensures that observations made every 16

days (e.g., on the 1st, 17th, and 33rd days of the year) have nearly identical VZA values. Consequently, annual observations for each pixel can be grouped into 16 distinct VZA categories, where the differences in VZA between groups are much greater than those within each group."

**Q10:** Figure 4: Google Earth is supposed to be capitalized. Please fix this in all locations (it is in other figures as well).

**Response:** Thanks for your constructive advice. We have corrected the capitalization of "Google Earth" in Figure 4, 6, 11, and throughout the entire text.

Figure 4: Spatial mismatch correction effectiveness in less-angular effect areas within airport cases. (a) Beijing International Airport, (b) Los Angeles International Airport, (c) Sydney International Airport. Google Earth images © 2025 Google LLC.

Figure 6: Angular effect and its correction effectiveness in different built-up environments. Google Earth images © 2025 Google LLC.

Figure 11. Comparison of angular correction results based on nadir correction (HDNTL) and annual mean correction (SFAC). Google Earth image © 2025 Google LLC.

Q11: "We also chose the runway pixel." - . should be "We also chose a runway pixel."

**Response:** Thanks for your insightful comment. We have revised the text.

Q12: In Figure 6, please write out what VZA stands for and the unit of measurement for each (in the scatterplot).

**Response:** Thanks for your constructive comment. We have corrected Figure 6.

Figure 6: Angular effect and its correction effectiveness in different built-up environments. Google Earth images © 2025 Google LLC.

**Q13:** Figure 7 also needs unit of analysis labels on the scatterplot. Please write out what RMSE and MAE are in the caption.

**Response:** Thanks for your comment. We have revised Figure 7 and its caption.

Figure 7: Interpolation accuracy test. (a) The Root Mean Square Error (RMSE) and Mean Absolute Error (MAE) of interpolation by different time windows, (b) The relative temporal weight of different time windows, (c) The comparison between reference data and its interpolation by the 5-day temporal window.